
# Holographic S-fold theories at one loop

Connor Behan

Mathematical Institute, University of Oxford, Andrew Wiles Building,
Radcliffe Observatory Quarter, Woodstock Road, Oxford, OX2 6GG, U.K.

## Abstract

A common feature of tree-level holography is that a correlator in one theory can serve as a generating function for correlators in another theory with less continuous symmetry. This is the case for a family of 4d CFTs with eight supercharges which have protected operators dual to gluons in the bulk. The most recent additions to this family were defined using S-folds which combine a spatial identification with an action of the S-duality group in type IIB string theory. Differences between these CFTs which have a dynamical origin first become manifest at one loop. To explore this phenomenon at the level of anomalous dimensions, we use the AdS unitarity method to bootstrap a one-loop double discontinuity. Compared to previous studies, the subsequent analysis is performed without any assumption about which special functions are allowed. Instead, the Casimir singular and Casimir regular terms are extracted iteratively in order to move from one Regge trajectory to the next. Our results show that anomalous dimensions in the presence of an S-fold are no longer rational functions of the spin.

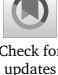

## Contents



# 1  Introduction

The AdS/CFT correspondence is a deep connection between two classification programs [1–3]. It allows one to explore the space of conformal field theories in $d$ dimensions by searching for low energy string backgrounds asymptotic to $AdS_{d+1}$. For any CFT constructed in this manner, it is possible in principle to go a step further and access the OPE data by computing correlation functions perturbatively. This expansion is organized by inverse powers of the central charge. Comparing early work such as [4–8] to the modern reformulation in [9–11] shows that a brute force application of this algorithm is extremely inefficient. A systematic analysis of holographic CFTs, made possible by bootstrap techniques [12–18], is therefore a recent endeavour. To complement the extensive results which are now available for $AdS_7 \times S^4$, $AdS_5 \times S^5$ and $AdS_4 \times S^7$ [19, 20], a growing body of work has focused on finding tree-level four-point functions for less supersymmetric backgrounds [21–24]. This paper will show that, without computing any new tree-level correlators, we can also gain a richer understanding of theory space beyond maximal supersymmetry by going to one loop.

Consider the stress tensor multiplet four-point function in 4d $\mathcal{N} = 4$ Super Yang Mills at tree-level. In the Mellin formalism [25–27], which we will review shortly, it takes the form

$$\mathcal{M}(s,t;\sigma,\tau) = \mathcal{M}_s(s,t;\sigma,\tau) + \tau^2 \mathcal{M}_s\left(t,s;\tfrac{\sigma}{\tau},\tfrac{1}{\tau}\right) + \sigma^2 \mathcal{M}_s\left(u,t;\tfrac{1}{\sigma},\tfrac{\tau}{\sigma}\right),$$

$$\mathcal{M}_s(s,t;\sigma,\tau) = -\frac{60}{c_T} \frac{(t-4)(u-4) + (t-4)(s+2)\sigma + (u-4)(s+2)\tau}{s-2}, \qquad (1.1)$$

where $\sigma$ and $\tau$ are invariant combinations of four polarization vectors $t_i$ associated with the $SO(6)$ R-symmetry. Importantly, the result (1.1) is the same for all three classical gauge groups

— all that changes between $SU(N)$, $SO(N)$ and $USp(2N)$ is the formula for $c_T$ in terms of $N$.[1] Indeed, the orientifold construction in [29] shows that one needs to do a loop calculation to be able to tell these theories apart. The flexibility of (1.1), however, does not stop there. After writing out the $\sigma$ and $\tau$ dependence, it is a simple exercise to make the replacement

$$t_i \cdot t_j \mapsto \frac{1}{2}\left(y_i \cdot \bar{y}_j + y_j \cdot \bar{y}_i\right),\tag{1.2}$$

where the $y_i$ and $\bar{y}_i$ are in the fundamental and anti-fundamental of $SU(3)$. Discarding the terms where any of these polarizations appear more than once yields another physical correlator at tree-level: the stress tensor multiplet four-point function of the holographic 4d $\mathcal{N} = 3$ SCFTs [30,31].[2] Along the same lines, one can also split (1.1) into components which transform irreducibly under flavour and R-symmetry groups of $SU(2)$. Among these is the stress tensor multiplet four-point function of two different $\mathcal{N} = 2$ SCFTs studied in [33,34] which are known to be planar equivalent to $\mathcal{N} = 4$ SYM.

The following sections will detail a loop calculation which "lifts the degeneracy" between a different collection of planar equivalent theories. These will be the 4d $\mathcal{N} = 2$ theories whose holographic correlators at order $1/c_J$ were computed in [24]. From this starting point, we will be able to propose general techniques while avoiding the technical complications which occur when some theories in the collection preserve more supercharges than others. In every other respect, the treatment will be able to serve as a blueprint for the $\mathcal{N} = 3$ example mentioned above which is based on an *S-fold*. Developed in [30,31], S-folds generalize the concept of an orientifold in type IIB string theory to include an action of the $SL(2,\mathbb{Z})$ duality group on the axio-dilaton. The S-folds we will use were introduced in [35] which showed that they can be made to preserve the same supercharges as the simple F-theory backgrounds of [36,37]. This leads to a 4d $\mathcal{N} = 2$ classification which complements the approach in [38–40].[3] These theories all contain a flavour current multiplet which can be regarded as a boundary mode for gluons in the bulk. Its four-point function at tree-level is given by

$$\mathcal{M}^{I_1 I_2 I_3 I_4}(s,t;\alpha) = \mathcal{M}_s^{I_1 I_2 I_3 I_4}(s,t;\alpha) + (\alpha-1)^2 \mathcal{M}_s^{I_3 I_2 I_1 I_4}\left(t,s;\tfrac{\alpha}{\alpha-1}\right) + \alpha^2 \mathcal{M}_s^{I_4 I_2 I_3 I_1}\left(u,t;\tfrac{1}{\alpha}\right),$$

$$\mathcal{M}_s^{I_1 I_2 I_3 I_4}(s,t;\alpha) = f^{I_1 I_2 J} f^{J I_3 I_4} \frac{6}{c_J} \frac{4-u+\alpha(t+u-8)}{s-2},\tag{1.3}$$

where $\alpha$ is a cross ratio for the $SU(2)$ R-symmetry and $f^{IJ}_{\ K}$ are the structure constants of the flavour group. The one-loop correction to (1.3) was recently computed in [44] yielding an expression from which one may extract double-trace operator dimensions. Our results will regard these anomalous dimensions as fundamental and show how they differ between three types of S-folds.

To actually compute these data, we will use the *AdS unitarity method* developed in [45].[4] This has a close connection to S-matrix theory which was made precise in [18]. Both in AdS and flat space, the absorptive part of a loop amplitude is determined by tree amplitudes which put the intermediate states on-shell. In reasonable holographic CFTs, these can be chosen from an infinite set of Kaluza-Klein modes. To deal with this mixing problem, [63–65] proposed a general algorithm and used it to analyze $\mathcal{N} = 4$ SYM. After further loop-level studies of $\mathcal{N} = 4$ SYM [66–70], the AdS unitarity method was first applied to other full-fledged theories

---

[1]For $USp(2N)$, one can make a choice between two gauge theories which have different line operators in the sense of [28].

[2]Keeping the quadratic terms instead would give us a four-point function of "extra supercurrent" multiplets meaning that the supersymmetry is automatically enhanced to $\mathcal{N} = 4$ [32].

[3]Further refinements, including a consideration of nonlocal operators, were made in [41–43].

[4]There is now a unifying picture [46] relating this to the split representation used in [47–51]. Alternative approaches to computing Witten diagrams with loops may be found in [52–62].

in [44, 71, 72].[5] Before extending this list, it is important to mention [75] which represents the current state of the art for $\mathcal{N} = 4$ SYM with gauge groups other than $SU(N)$. In this case, one-loop anomalous dimensions cease to be rational functions of the spin — a property which appears to be shared by S-folds.

This paper is organized as follows. Section 2 discusses the theories of [35] and sets up standard tools for their analysis such as the superconformal block expansion, superconformal Ward identity and Lorentzian inversion formula. Section 3 then reviews what is known at the disconnected and tree levels [24] where the effects of an S-fold are purely kinematical. Some examples of OPE inversion provide preparation for the longer calculations at one loop. We begin working at one loop in section 4 which includes the debut of our method for isolating operator content in a model independent way. The main results are a handful of mixed and unmixed anomalous dimensions. A few of these have fixed spin and the rest are given as asymptotic expansions around large spin. After the conclusion in section 5, some technical points are developed further in the appendices.

## 2 Setup

In this section, we describe the prerequisites for writing down specific four-point functions. Section 2.1 discusses the $AdS_5$ geometries which give rise to our theories of interest. The Kaluza-Klein modes found in [36, 37] turn out to transform differently in the presence of an S-fold and we give a simple rule for determining which states are projected out. Section 2.2 establishes our conventions for superconformal correlators, while section 2.3 reviews the Lorentzian inversion formula [18]. In both cases we focus on external operators with equal scaling dimensions but otherwise give general results. In section 2.4, we derive a group theoretic prescription which shows how OPE coefficients with an S-fold are related to those without.

### 2.1 S-fold backgrounds

One way to construct various rank $N$ CFTs, including those of Argyres-Douglas and Minahan-Nemeschansky type, is to consider the system

$$\overbrace{\underbrace{\mathbb{R}^{1,3} \times \mathbb{C}_1 \times \mathbb{C}_2}_{7} \times \mathbb{C}_3}^{D3} \tag{2.1}$$

of $N$ D3 branes coincident with a 7-brane of F-theory.[6] The 7-brane must be chosen so that its worldvolume harbors one of the gauge groups in table 1. These all give rise to a deficit angle in the transverse $\mathbb{C}_3$ and all but one of them lead to a fixed (strongly coupled) value for the axio-dilaton $\tau$.

### 2.1.1 Isometries and gauge symmetry

When $N$ is large, the near-horizon metric is given by

$$ds^2 = ds^2_{AdS_5} + d\phi^2 + \left(\frac{2-\nu}{2}\right)^2 \cos^2\phi \, d\theta^2 + \sin^2\phi \, ds^2_{S^3}, \tag{2.2}$$

---

[5]Line defects of CFTs have provided another interesting application [73, 74].

[6]These are bound states of D7 branes (on which strings can end) along with certain images of D7 branes under $SL(2, \mathbb{Z})$ transformations which make them non-perturbative objects.

Table 1: Basic data for the 4d SCFTs probing F-theory singularities. The 7-brane tension creates a deficit angle of $\nu\pi$ and their monodromies fix the value of the axio-dilaton $\tau$. These possibilities are also commonly labelled by Kodaira singular fibers but that will not play a role here.

| $G$ | $A_0$ | $A_1$ | $A_2$ | $D_4$ | $E_6$ | $E_7$ | $E_8$ |
|---|---|---|---|---|---|---|---|
| $\nu$ | $1/3$ | $1/2$ | $2/3$ | $1$ | $4/3$ | $3/2$ | $5/3$ |
| $\tau$ | $e^{\frac{\pi i}{3}}$ | $e^{\frac{\pi i}{2}}$ | $e^{\frac{\pi i}{3}}$ | — | $e^{\frac{\pi i}{3}}$ | $e^{\frac{\pi i}{2}}$ | $e^{\frac{\pi i}{3}}$ |

which makes it clear that the 7-branes, wrapping an $S^3 \subset S^5$, have broken the isometries according to

$$SO(6) \to SU(2)_L \times SU(2)_R \times U(1)_R . \tag{2.3}$$

The single particle spectrum for this background is composed of two types of Kaluza-Klein modes: those of 10d IIB SUGRA reduced on the internal manifold of (2.2) and those of 8d $G$ SYM reduced on $S^3$. As in [24], we will only consider correlators where all external operators are of the second type.[7] These transform in the adjoint representation of the gauge group with generators denoted by $T_I$. In this holographic setup, the 7-brane gauge group is a flavour group of the CFT which we write as $G_F = G$. Following [37], let us focus on the internal components of a generic gauge field $A_\mu(x, y) = A_\mu^I(x, y)T_I$. Reducing it on $S^{n-1}$ means writing the spherical harmonic expansion

$$A_a^I(x, y) = \sum_{\mathfrak{M}} A_{\mathfrak{M}}^I(x) Y_a^{\mathfrak{M}}(y), \tag{2.4}$$

where $\mathfrak{M}$ is a Gelfand-Tsetlin pattern of the $SO(n)$ representation given by the simplest hook diagram. This is because each basis element for the space of tensors having this index symmetry yields a vector spherical harmonic obtained by pulling back

$$c_a^{b_1 \dots b_{p-1}} x_{b_1} \dots x_{b_{p-1}}, \quad x \in \mathbb{R}^n \tag{2.5}$$

to $S^{n-1}$. When $n = 4$, these representations are of course reducible. In terms of the $SU(2)$ spins $(j_L, j_R)$, they can be written $\left(\frac{p}{2}, \frac{p-2}{2}\right) \oplus \left(\frac{p-2}{2}, \frac{p}{2}\right)$ for $p \geq 2$. Although these states are on equal footing at the level of bosonic symmetries, only one of them can be a superconformal primary. This will be the one with a spin of $\frac{p}{2}$ under the $SU(2)$ that rotates the supercharges and a spin of $\frac{p-2}{2}$ under the $SU(2)$ that commutes with the supercharges [37]. As suggested by the notation, we choose these to be $SU(2)_R$ and $SU(2)_L$ respectively. Several naming schemes have been used for these theories, most recently $\mathcal{S}_{G,1}^{(N)}$.

### 2.1.2 Enter S-folds

The idea of [35] was to have the $z_1 = z_2 = z_3 = 0$ locus within the 7-branes coincide with the fixed plane of an S-fold which preserves $\mathcal{N} = 3$. Its action is

$$(z_1, z_2, z_3) \mapsto \left(e^{\frac{2\pi i}{k}} z_1, e^{-\frac{2\pi i}{k}} z_2, e^{\frac{2\pi i}{k}} z_3\right), \quad \tau \mapsto g\,\tau, \tag{2.6}$$

where $g \in SL(2, \mathbb{Z})$ has order $k$. This immediately implies that there are only four non-trivial values of $k$. Determining the allowed 7-branes is not as simple as comparing tables 1 and 2. Due to the deficit angle, it is really $z_3^{\frac{2-\nu}{2}}$ which forms a good co-ordinate on the transverse space. One should therefore consider S-folds with the subgroups $\mathbb{Z}_{\frac{2k}{2-\nu}}$ and check that they

---

[7]With this choice, the effects of the deficit angle on operator dimensions will not be visible until higher orders.

Table 2: S-folds are labelled by $k$ such that $\mathbb{Z}_k$ is a subgroup of $SL(2,\mathbb{Z})$. In each case, we give a possible generator and the value of $\tau$ that it fixes.

| $k$ | 2 | 3 | 4 | 6 |
|---|---|---|---|---|
| $g$ | $-I$ | $(ST)^2$ | $S$ | $ST$ |
| $\tau$ | — | $e^{\frac{\pi i}{3}}$ | $e^{\frac{\pi i}{2}}$ | $e^{\frac{\pi i}{3}}$ |

Table 3: Flavour symmetries and central charges for the twelve known classes of 4d $\mathcal{N} = 2$ S-folds. See [41, 43] for more information including Coulomb branch scaling dimensions (which differ between the $\mathcal{S}_{G,k}^{(N)}$ and $\mathcal{T}_{G,k}^{(N)}$ theories) and a specific breakdown of how many units of central charge are due to each factor of $G_F$.

| | $G_F$ | $c_J$ | $c_T$ |
|---|---|---|---|
| $\mathcal{S}_{A_2,2}^{(N)}$ | $C_1 \times A_0$ | $\frac{3}{2}(3N+1)$ | $10(9N^2+9N+1)$ |
| $\mathcal{S}_{D_4,2}^{(N)}$ | $C_2 \times A_1$ | $\frac{3}{2}(12N+1)$ | $10(12N^2+15N+2)$ |
| $\mathcal{S}_{E_6,2}^{(N)}$ | $C_4$ | $\frac{3}{2}(6N+1)$ | $10(18N^2+27N+4)$ |
| $\mathcal{T}_{A_2,2}^{(N)}$ | $A_2$ | $\frac{9N}{2}$ | $90N^2$ |
| $\mathcal{T}_{D_4,2}^{(N)}$ | $B_3$ | $6N$ | $30N(4N+1)$ |
| $\mathcal{T}_{E_6,2}^{(N)}$ | $F_4$ | $9N$ | $90N(2N+1)$ |
| $\mathcal{S}_{A_1,3}^{(N)}$ | $A_0$ | $0$ | $10(12N^2+11N+1)$ |
| $\mathcal{S}_{D_4,3}^{(N)}$ | $A_2$ | $3(6N+1)$ | $30(6N^2+7N+1)$ |
| $\mathcal{T}_{A_1,3}^{(N)}$ | $A_1$ | $4N$ | $10N(12N-5)$ |
| $\mathcal{T}_{D_4,3}^{(N)}$ | $G_2$ | $6N$ | $30N(6N-1)$ |
| $\mathcal{S}_{A_2,4}^{(N)}$ | $A_1$ | $3(6N+1)$ | $20(9N^2+9N+1)$ |
| $\mathcal{T}_{A_2,4}^{(N)}$ | $A_1$ | $\frac{9N}{2}$ | $90N(2N-1)$ |

lead to compatible values of $\tau$ [35]. This yields six possible backgrounds $\mathcal{S}_{G,k}^{(N)}$ for D3 branes to probe which are $\mathcal{S}_{A_2,2}^{(N)}$, $\mathcal{S}_{D_4,2}^{(N)}$, $\mathcal{S}_{E_6,2}^{(N)}$, $\mathcal{S}_{A_1,3}^{(N)}$, $\mathcal{S}_{D_4,3}^{(N)}$ and $\mathcal{S}_{A_2,4}^{(N)}$. As found by [41], there is a related S-fold theory $\mathcal{T}_{G,k}^{(N)}$ which may be obtained from $\mathcal{S}_{G,k}^{(N)}$ by a partial Higgsing. In both cases, central charges as a function of $N$ were obtained by the method of [76]. Table 3 lists them in conventions such that

$$\left\langle T_{\mu\nu}(x)T_{\rho\sigma}(0)\right\rangle = \frac{c_T}{6\pi^4}\left(I_{\mu\sigma}I_{\nu\rho}+I_{\mu\rho}I_{\nu\sigma}-\frac{1}{2}\delta_{\mu\nu}\delta_{\rho\sigma}\right)\frac{1}{x^8}$$

$$\left\langle J_\mu^I(x)J_\nu^J(0)\right\rangle = \frac{c_J}{2\pi^2}\frac{I_{\mu\nu}\delta^{IJ}}{x^6}\,, \tag{2.7}$$

where $I_{\mu\nu} \equiv \delta_{\mu\nu} - 2\frac{x_\mu x_\nu}{x^2}$.

Taking the near-horizon limit again, (2.2) becomes

$$ds^2 = ds_{AdS_5}^2 + d\phi^2 + \left(\frac{2-\nu}{2k}\right)^2\cos^2\phi\,d\theta^2 + \sin^2\phi\,ds_{S^3/\mathbb{Z}_k}^2. \tag{2.8}$$

Using Hopf co-ordinates, we can parametrize the equatorial $S^3$ at $\phi = \frac{\pi}{2}$ by

$$z_1 \equiv x_3 + ix_4 = \cos\beta e^{i\omega}\,, \quad z_2 \equiv x_1 + ix_2 = \sin\beta e^{i\tilde{\omega}}\,, \tag{2.9}$$

where $(\omega, \tilde{\omega}) \sim \left(\omega + \frac{2\pi}{k}, \tilde{\omega} - \frac{2\pi}{k}\right)$ by (2.6). Now consider what happens to the spherical harmonics for superconformal primaries. Defining $\sigma^\mu = (\vec{\sigma}, iI)$, (2.5) may be written in the

convenient form

$$(j_L, j_R) = \left( \frac{p-2}{2}, \frac{p}{2} \right) \qquad c^{\alpha_1 \dots \alpha_{p-2}; \bar{\alpha}_1 \dots \bar{\alpha}_p} \mathrm{x}_{\alpha_1 \bar{\alpha}_1} \dots \mathrm{x}_{\alpha_{p-2} \bar{\alpha}_{p-2}}, \quad \mathrm{x}_{\alpha \bar{\alpha}} = x_\mu \sigma^\mu_{\alpha \bar{\alpha}}. \tag{2.10}$$

To complete the Gelfand-Tsetlin pattern we need $m_L$ which is half the difference between the number of unbarred $\pm$ indices and $m_R$ which is half the difference between the number of barred $\pm$ indices. These are of course the Cartans of $U(1)$ satisfying $|m_L| \leq j_L$ and $|m_R| \leq j_R$. Plugging (2.9) into (2.10), we therefore see that the associated spherical harmonics have $m_R$ powers of $e^{i(\omega + \tilde{\omega})}$ and $m_L$ powers of $e^{i(\omega - \tilde{\omega})}$. Demanding that the latter be invariant under the S-fold action leads to the main result of this subsection. *A state is part of the Kaluza-Klein spectrum if and only if $2m_L$ is divisible by $k$.* Note that when $k = 2$, this is simply a restriction to integer values of $m_L$ which we can implement by restricting to integer values of $j_L$. We therefore find, in agreement with the results of [35], that $SU(2)_L$ is preserved for $k = 2$ and breaks to $U(1)_L$ for all other values of $k$.[8]

Before moving on, it is interesting to note that $G_F \neq G$ for most of the rows in table 3. In other words, $G$ and $SU(2)_L$ are both broken to an invariant subgroup by the S-fold. For $\mathcal{S}_{G,k}^{(N)}$ theories, [35] indirectly identified the subgroups of $G$ which commute with the asymptotic holonomy around the D3 branes. It was later pointed out in [43] that the classification of automorphisms of $G$ which cannot be undone by a gauge transformation restricts these invariant subgroups to a very short list. To derive them microscopically, one should presumably keep track of how the constituents of a 7-brane in F-theory are transformed into each other by $SL(2, \mathbb{Z})$. Their electric and magnetic charges are usually taken to be

$$A : (1, 0) \qquad B : (3, 1) \qquad C : (1, 1), \tag{2.11}$$

without loss of generality. The branes labelled $A$ are ordinary D7 branes which are magnetic sources for the zero form in the axio-dilaton. There have been at least two proposals for how Chan-Paton factors of the individual $(p, q)$ branes serve to build up the gauge group. One uses open strings which wind around non-perturbative branes in judiciously chosen ways [77] and another uses multi-pronged open strings [78]. Fortunately, the holographic calculations that follow can be done without committing to any particular flavour group. We will therefore refer to $G_F$ abstractly where $|G_F|$ will be its dimension and $G_F^\vee$ the product of the dual Coxeter number and the length of the longest root.

## 2.2 Superconformal kinematics

The superconformal primaries dual to single-particle states on the singular locus belong to $\frac{1}{2}$-BPS multiplets of the 4d $\mathcal{N} = 2$ superconformal algebra. These were denoted by $B\bar{B}[0; 0]_p^{(p,0)}$ in [79]. Proving this is a simple exercise in representation theory. The SYM fields on the 7-brane have Lorentz spin at most 1 which makes any other multiplet too long to fit.[9] The quantum numbers of these operators are

$$\Delta = p, \quad \ell = 0, \quad j_L = \frac{p-2}{2}, \quad j_R = \frac{p}{2}, \tag{2.12}$$

along with the adjoint representation of $G_F$. Knowing this, it is convenient to use the notation

$$\mathcal{O}_j^I(x; v, \bar{v}) \equiv \mathcal{O}_{\alpha_1 \dots \alpha_{2j}; \bar{\alpha}_1 \dots \bar{\alpha}_{2j+2}}^I(x) v^{\alpha_1} \dots v^{\alpha_{2j}} \bar{v}^{\bar{\alpha}_1} \dots \bar{v}^{\bar{\alpha}_{2j+2}} \tag{2.13}$$

---

[8]This compares nicely to the $AdS_5 \times S^5$ story because the $k = 2$ S-fold of $\mathcal{N} = 4$ SYM reduces to the standard orientifold [29].

[9]This is entirely analogous to how SURGRA fields of spin at most 2 limit us to $\frac{1}{2}$-BPS multiplets of 4d $\mathcal{N} = 4$. The only way out would be to imagine that higher-spin descendants of single-trace operators are double-trace — an equation of motion which would lead to contradictions.

for the $k = 2$ S-folds.[10] For the $k = 3$ and $k = 4$ S-folds, we will instead refer to individual components $\mathcal{O}^I_{j,m}(x; \bar{v})$ where $m$ is the $U(1)_L$ charge. A four-point function of the operators (2.13) will involve the cross ratios

$$U \equiv z\bar{z} = \frac{x^2_{12}x^2_{34}}{x^2_{13}x^2_{24}}, \quad V \equiv (1-z)(1-\bar{z}) = \frac{x^2_{14}x^2_{23}}{x^2_{13}x^2_{24}}, \quad \alpha = \frac{\bar{v}_{13}\bar{v}_{24}}{\bar{v}_{12}\bar{v}_{34}}, \quad \beta = \frac{v_{13}v_{24}}{v_{12}v_{34}}, \quad (2.14)$$

where

$$x_{ij} = x_i - x_j, \quad v_{ij} = v^\alpha_i v^\beta_j \epsilon_{\alpha\beta}, \quad \bar{v}_{ij} \bar{v}^{\bar{\alpha}}_i \bar{v}^{\bar{\beta}}_j \epsilon_{\bar{\alpha}\bar{\beta}}. \quad (2.15)$$

It is possible to make the dynamical part a degree $\mathcal{E}$ polynomial in $\alpha$ and a degree $\mathcal{E} - 2$ polynomial in $\beta$ where

$$\mathcal{E} = \begin{cases} 2j_{\min} + 2, & 2(j_{\min} + j_{\max}) \leq \sum^4_{i=1} j_i, \\ \sum^4_{i=1} j_i - 2j_{\max} + 2, & 2(j_{\min} + j_{\max}) > \sum^4_{i=1} j_i, \end{cases} \quad (2.16)$$

is the *extremality*. This is done by extracting a kinematic prefactor of the form

$$\left\langle \mathcal{O}^{I_1}_{j_1}(x_1; v_1, \bar{v}_1) \mathcal{O}^{I_2}_{j_1}(x_2; v_2, \bar{v}_2) \mathcal{O}^{I_3}_{j_3}(x_3; v_3, \bar{v}_3) \mathcal{O}^{I_4}_{j_4}(x_4; v_4, \bar{v}_4) \right\rangle = G^{I_1 I_2 I_3 I_4}(U, V; \alpha, \beta)$$

$$\times \prod_{i<j} \left( \frac{v_{ij}\bar{v}_{ij}}{x^2_{ij}} \right)^{\gamma_{ij}} \left( \frac{\bar{v}_{12}\bar{v}_{34}}{x^2_{12}x^2_{34}} \right)^{\mathcal{E}} (v_{12}v_{34})^{\mathcal{E}-2}. \quad (2.17)$$

The exponents are given by

$$\gamma_{12} = 0, \qquad \gamma_{14} = 2j_1 + 2 - \mathcal{E}, \qquad \gamma_{13} = 0,$$
$$\gamma_{24} = j_2 + j_4 - j_1 - j_3, \ \gamma_{23} = j_1 + j_2 + j_3 - j_4 - \mathcal{E} + 2, \ \gamma_{34} = j_4 + j_3 - j_2 - j_1 \quad (2.18)$$

when $j_1 \leq j_2 \leq j_3 \leq j_4$ and they are determined by a straightforward permutation otherwise.[11]

Expanding a generic four-point function into superconformal blocks is often a difficult task. With $\frac{1}{2}$-BPS four-point functions, one is able to take an alternative route: using a solution of the superconformal Ward identity to write (2.17) in terms of two auxiliary functions which are easier to expand.[12] The relevant superconformal Ward identity, first studied in [81, 82], takes the simple form

$$\left[ z\frac{\partial}{\partial z} - \alpha\frac{\partial}{\partial \alpha} \right] G(z, \bar{z}; \alpha, \beta) \bigg|_{\alpha = z^{-1}} = 0, \quad (2.19)$$

where we have suppressed the four adjoint indices. Its solution is given by

$$G(z, \bar{z}; \alpha, \beta) = \frac{z(1 - \alpha\bar{z})f(\bar{z}; \beta) - \bar{z}(1 - \alpha z)f(z; \beta)}{z - \bar{z}} + (1 - \alpha z)(1 - \alpha\bar{z})H(z, \bar{z}; \alpha, \beta), \quad (2.20)$$

where

$$f(z; \beta) = G(z, \bar{z}; \bar{z}^{-1}, \beta), \quad f(\bar{z}; \beta) = G(z, \bar{z}; z^{-1}, \beta) \quad (2.21)$$

and $H(z, \bar{z}; \alpha, \beta)$ (now of degree $\mathcal{E} - 2$ in both $\alpha$ and $\beta$) is defined by subtraction. Similar decompositions exist in 6d theories as well [81].[13] Contributions to (2.20) come in three

---

[10]We have reversed the bars with respect to [24] since we will be focusing on $SU(2)_L$ more than $SU(2)_R$.

[11]This notation might be a little *too* compact. When handling $j_2 \leq j_1 \leq j_3 \leq j_4$ for instance, $\gamma_{14}$ becomes $2j_2 + 2 - \mathcal{E}$ which is not $\gamma_{24}$.

[12]This is what [80] referred to as the expansion in atomic blocks.

[13]Auxiliary correlators are probably responsible for the biggest difference in look and feel between the even and odd dimensional superconformal bootstrap literature. See *e.g.* the need to remove redundant crossing equations in [83].

Table 4: Expressions for the superconformal blocks in a form which shows that they manifestly satisfy the superconformal Ward identity. One can see that the $j = 0, 1$ elements of the first row (identity and moment map) only contribute to the chiral correlator. The $j = 0$ multiplets in the second row would as well but these are ruled out since they contain higher-spin currents.

| Multiplet | $f(z)$ part | $H(U, V; \alpha)$ part |
|---|---|---|
| $B\bar{B}[0; 0]_{2j}^{(2j,0)}$ | $k_j(z)$ | $\sum_{k=0}^{j-2} U^{-1} g_{j+k+2, j-k-2}(U, V) \mathcal{Y}_k(\alpha)$ |
| $A\bar{A}[\ell; \ell]_{2j+\ell+2}^{(2j,0)}$ | $k_{j+\ell+2}(z)$ | $-\sum_{k=0}^{j-1} U^{-1} g_{j+\ell+k+4, j+\ell-k}(U, V) \mathcal{Y}_k(\alpha)$ |
| $L\bar{L}[\ell; \ell]_{\Delta}^{(2j,0)}$ | $0$ | $U^{-1} g_{\Delta+2, \ell}(U, V) \mathcal{Y}_j(\alpha)$ |

types according to the selection rule

$$
B\bar{B}[0; 0]_{2j_1}^{(2j_1,0)} \otimes B\bar{B}[0; 0]_{2j_2}^{(2j_2,0)} = \bigoplus_{j=|j_{12}|}^{j_1+j_2} B\bar{B}[0; 0]_{2j}^{(2j,0)} \tag{2.22}
$$

$$
\oplus \bigoplus_{\ell=0}^{\infty} \left[ \bigoplus_{j=|j_{12}|}^{j_1+j_2-1} A\bar{A}[\ell; \ell]_{2j+\ell+2}^{(2j,0)} \oplus \bigoplus_{j=|j_{12}|}^{j_1+j_2-2} L\bar{L}[\ell; \ell]_{\Delta}^{(2j,0)} \right] .
$$

They can be expressed in terms of the 1d bosonic blocks

$$
k_h(z) = z^h {}_2F_1(h, h; 2h; z), \tag{2.23}
$$

4d bosonic blocks

$$
g_{\Delta,\ell}(z, \bar{z}) = \frac{z\bar{z}}{z - \bar{z}} \left[ k_{\frac{\Delta+\ell}{2}}(z) k_{\frac{\Delta-\ell-2}{2}}(\bar{z}) - (z \leftrightarrow \bar{z}) \right] \tag{2.24}
$$

and $SU(2)$ polynomials

$$
\mathcal{Y}_j(\alpha) = k_{-j}(\alpha^{-1}) = \frac{j!^2}{(2j)!} P_j(2\alpha - 1). \tag{2.25}
$$

The procedure for working them out, first explained in [84], leads to table 4. We have checked that these results agree with the 4d case of the dimensionally continued blocks in [85]. In general one can examine the structure of a superconformal multiplet most easily using $G(z, \bar{z}; \alpha, \beta)$ while certain mysterious features of holographic correlators (*e.g.* hidden conformal symmetry [86] and a double copy relation [87]) are most transparently encoded in $H(z, \bar{z}; \alpha, \beta)$. We have chosen to use the latter in order to simplify the process of inverting the OPE.

## 2.3 The Lorentzian inversion formula

We now turn to the result of [18] which captures a remarkable fact about Regge bounded four-point functions — all of their OPE data for sufficiently large spin can be extracted from a simpler piece called the *double discontinuity*. The double discontinuity, which can also be seen as a double commutator, is given by

$$
\text{dDisc}[G(z, \bar{z})] = G(z, \bar{z}) - \frac{1}{2} G^{\circlearrowright}(z, \bar{z}) - \frac{1}{2} G^{\circlearrowleft}(z, \bar{z}) \tag{2.26}
$$

in the case of external operators (which could all be different) having pairwise equal scaling dimensions. The arrows are shorthand for clockwise and counter-clockwise continuations around a branch point in $\bar{z}$. As usual, we will take $\bar{z} = 1$ so that dDisc refers to the $t$-channel

double discontinuity everywhere in this paper.[14] To start, we will write the Lorentzian inversion formula for the spectral density as

$$c(\Delta, \ell) = \frac{1}{4} \kappa_{\frac{\Delta+\ell}{2}} \int_0^1 \frac{dz\, d\bar{z}}{z^2\, \bar{z}^2} \left(\frac{z-\bar{z}}{z\bar{z}}\right)^2 g_{\ell+3, \Delta-3}(z, \bar{z}) \mathrm{dDisc}[G(z, \bar{z}) + (-1)^\ell \widehat{G}(z, \bar{z})], \quad (2.27)$$

where $\kappa_h = \frac{\Gamma(h)^4}{2\pi^2 \Gamma(2h) \Gamma(2h-1)}$. In the present context, $G(z, \bar{z})$ should be viewed as the dynamical part of $\left\langle \mathcal{O}_j^{I_1} \mathcal{O}_j^{I_2} \mathcal{O}_j^{I_3} \mathcal{O}_j^{I_4} \right\rangle$ projected onto a given index structure, with $\widehat{G}(z, \bar{z})$ being the dynamical part of $\left\langle \mathcal{O}_j^{I_2} \mathcal{O}_j^{I_1} \mathcal{O}_j^{I_3} \mathcal{O}_j^{I_4} \right\rangle$ projected onto the same structure. The two will differ by a sign which sometimes enhances $(-1)^\ell$ and sometimes cancels it.

As written, (2.27) is only a partial solution to the problem of decomposing a superconformal four-point function. After using it to read off the coefficients of conformal blocks, they would need to be reassembled to account for superconformal blocks through a series of recurrence relations. It is therefore desirable to use the auxiliary correlator of an $\mathcal{N} = 2$ theory where the coefficient of an ordinary conformal block already captures the contribution of an entire long multiplet. For external operators that transform under $SU(2)_L$, the projections mentioned above are

$$H_a(z, \bar{z}; j_L, j_R) = \oint \frac{d\alpha}{2\pi i} k_{j_R+1}(\alpha^{-1}) \oint \frac{d\beta}{2\pi i} k_{j_L+1}(\beta^{-1}) P_a^{I_1 I_2 | I_3 I_4} H^{I_1 I_2 I_3 I_4}(z, \bar{z}; \alpha, \beta),$$

$$\widehat{H}_a(z, \bar{z}; j_L, j_R) = (-1)^{j_L + j_R + |\mathbf{R}_a|} H_a(z, \bar{z}; j_L, j_R). \quad (2.28)$$

The last line assumes that the first two and last two operators are respectively in the same representation of $SU(2)_L \times SU(2)_R \times G_F$.[15] In the $G_F$ projectors used to write (2.28), $a$ runs over all representations that can appear in the tensor product of two adjoints. Their properties have been discussed in [21, 24] and we recall

$$P_a^{I_1 I_2 | I_3 I_4} = P^{I_3 I_4 | I_1 I_2}, \quad P_a^{I_1 I_2 | I_3 I_4} P_b^{I_1 I_2 | I_3 I_4} = \delta_{ab} \dim(\mathbf{R}_a),$$

$$P_a^{I_1 I_2 | I_3 I_4} = (-1)^{|\mathbf{R}_a|} P_a^{I_2 I_1 | I_3 I_4}, \quad P_a^{I_1 I_2 | I_3 I_4} P_b^{I_4 I_3 | I_5 I_6} = \delta_{ab} P^{I_1 I_2 | I_5 I_6}. \quad (2.29)$$

Following [66, 86], it is convenient to switch to the variables

$$h = \frac{\Delta - \ell}{2}, \qquad \bar{h} = \frac{\Delta + \ell + 2}{2}, \quad (2.30)$$

so that the inversion formula factorizes upon plugging in the explicit form of (2.24). In terms of (2.30) and the quantity $r_h = \frac{\Gamma(h)^2}{\Gamma(2h-1)}$,

$$c_a(h, \bar{h}; j_L, j_R) = -\frac{r_{\bar{h}}^2}{4\pi^2} \int_0^1 \frac{dz}{z^2} k_{1-h}(z) \int_0^1 \frac{d\bar{z}}{\bar{z}^2} \frac{k_{\bar{h}}(\bar{z})}{\bar{h} - \frac{1}{2}} \mathrm{dDisc}[(z - \bar{z}) H_a(z, \bar{z}; j_L, j_R)], \quad (2.31)$$

is the spectral density which gives the coefficients of $g_{\Delta+2, \ell}(U, V)$ blocks in $UH_a(U, V; j_L, j_R)$.[16] A full trajectory's worth may be computed by isolating a single pole in $h$. When this is done, all operators with this twist have their OPE coefficients determined by the residue with $\bar{h}$ giving the spin dependence.

---

[14]The $s$-channel OPE data comes from separate inversion integrals for the other two channels, *i.e.* one for $\bar{z} = 1$ and the other for $\bar{z} = \infty$. The usual step is to rewrite the $u$-channel double discontinuity in terms of the $t$-channel by permuting the first two operators. This requires the four-point function in question to satisfy crossing.

[15]Note that only integer values of $j_L$ and $j_R$ can be exchanged in an OPE of (2.13) operators that have the same scaling dimensions.

[16]Clearly, we have assumed that $\ell + j_L + j_R + |\mathbf{R}_a|$ is even. If it were odd, (2.31) would be zero.

To see that anomalous dimensions arise from higher poles in $h$, it is easy to expand

$$c(\Delta, \ell) \sim -\frac{\sum_{l=0}^{\infty} c_J^{-l} a^{(l)}}{\Delta - \Delta^{(0)} - \sum_{l=1}^{\infty} c_J^{-l} \gamma^{(l)}} \, . \tag{2.32}$$

Keeping only the terms applicable to a one-loop calculation leads to

$$
\begin{aligned}
-c^{(0)}(\Delta, \ell) &\sim \frac{a^{(0)}}{\Delta - \Delta^{(0)}} \, , \\
-c^{(1)}(\Delta, \ell) &\sim \frac{a^{(1)}}{\Delta - \Delta^{(0)}} + \frac{a^{(0)} \gamma^{(1)}}{(\Delta - \Delta^{(0)})^2} \, , \\
-c^{(2)}(\Delta, \ell) &\sim \frac{a^{(2)}}{\Delta - \Delta^{(0)}} + \frac{a^{(0)} \gamma^{(2)} + a^{(1)} \gamma^{(1)}}{(\Delta - \Delta^{(0)})^2} + \frac{a^{(0)} \gamma^{(1)2}}{(\Delta - \Delta^{(0)})^3} \, .
\end{aligned}
\tag{2.33}
$$

For our purposes, these are all *averaged* quantities since the double-trace operators which receive anomalous dimensions are highly degenerate. Starting from the double-trace value $h = 2j + n + 2 + \delta$, we can consider the poles as $\delta \to 0$. Comparing to (2.33) yields

$$c_a(2j + n + 2 + \delta, \bar{h}; j_L, j_R) \sim -\sum_{l=0}^{\infty} \frac{1}{c_J^l} \sum_{m=0}^{l} \frac{S_a^{(l,m)}(n, \bar{h}; j_L, j_R)}{(2\delta)^{m+1}} \, , \tag{2.34}$$

$$\left\langle a^{(0)} \right\rangle = S^{(0,0)}, \quad \left\langle a^{(1)} \right\rangle = S^{(1,0)} + \frac{1}{2} \frac{\partial}{\partial \bar{h}} S^{(1,1)}, \quad \left\langle a^{(0)} \gamma^{(2)} + a^{(1)} \gamma^{(1)} \right\rangle = S^{(2,1)} + \frac{1}{2} \frac{\partial}{\partial \bar{h}} S^{(2,2)},$$

$$\left\langle a^{(0)} \gamma^{(1)} \right\rangle = S^{(1,1)}, \quad \left\langle a^{(0)} \gamma^{(1)2} \right\rangle = S^{(2,2)}, \quad \left\langle a^{(2)} \right\rangle = S^{(2,0)} + \frac{1}{2} \frac{\partial}{\partial \bar{h}} S^{(2,1)} + \frac{1}{8} \frac{\partial^2}{\partial \bar{h}^2} S^{(2,2)} \, .$$

The derivatives appear because $\bar{h}$ differs from $h$ by an integer and therefore depends on $\delta$ as well. More information on the specific integrals we will use to determine (2.34) is given in Appendix A.

## 2.4 Projected OPE coefficients

The above advocates for an abstract definition of a 4d $\mathcal{N} = 2$ S-fold as a restriction to single-trace operators with $2m_L | k$. Although this changes the global symmetry of the theory, the couplings at leading order are still fixed by kinematics. Kaluza-Klein modes interact just as they would inside $SU(2)_L$ multiplets in analogy with how low energy scattering cannot tell which other parts of the spectrum have been made heavy. To see what this means for correlation functions, consider a two-point function with only the $SU(2)_L$ dependence written out.

$$\left\langle \mathcal{O}_j(v_1) \mathcal{O}_j(v_2) \right\rangle = v_{12}^{2j} = \sum_{m=-j}^{j} \binom{2j}{j+m} (v_1^+ v_2^-)^{j+m} (-v_1^- v_2^+)^{j-m} \, . \tag{2.35}$$

Each term in the sum corresponds to a different $U(1)_L$ two-point function.[17] We can make them have the unit normalization

$$\left\langle \mathcal{O}_{j,m} \mathcal{O}_{j,-m} \right\rangle = (-1)^{j-m}, \tag{2.36}$$

by forcing the polarization components to take the values

$$v_i^{\pm} \mapsto \binom{2j_i}{j_i + m_i}^{-\frac{1}{4j_i}} \, . \tag{2.37}$$

---

[17]Suppressing other quantum numbers might make it look like these operators are fermionic. If $j_L$ is a half-integer then $j_R$ is as well which cancels the anti-symmetry of (2.35). The apparent factor of $(-1)^{2m}$ upon switching the two operators in (2.36) is innocuous for the same reason.

In a similar way,

$$\langle \mathcal{O}_{j_1}(v_1)\mathcal{O}_{j_2}(v_2)\mathcal{O}_{j_3}(v_3)\rangle = C_{j_1,j_2,j_3}\, v_{12}^{j_1+j_2-j_3} v_{23}^{j_2+j_3-j_1} v_{31}^{j_3+j_1-j_2} \tag{2.38}$$

can be projected onto a set of $U(1)_L$ charges that sum to zero. According to the Wigner-Eckart theorem, the answer will be related to a Clebsch-Gordan coefficient by a factor which only depends on $j_1$, $j_2$ and $j_3$. We will verify this (and determine the overall factor) by an explicit calculation.

The first step is to write each $v_{ij}$ power law as a sum with the form of (2.35). Each summation index, which we call $m_{ij}$, gives $v_i^{\pm}$ a charge of $m_{ij}$ and $v_j^{\pm}$ a charge of $-m_{ij}$. The coefficient adorning them is

$$\binom{j_1+j_2-j_3}{\frac{j_1+j_2-j_3}{2}+m_{12}}\binom{j_2+j_3-j_1}{\frac{j_2+j_3-j_1}{2}+m_{23}}\binom{j_3+j_1-j_2}{\frac{j_3+j_1-j_2}{2}+m_{31}}(-1)^{\frac{j_1+j_2+j_3}{2}-m_{12}-m_{23}-m_{31}}. \tag{2.39}$$

To fix the charge in each position, the sum must be restricted to the terms which satisfy

$$m_{12}-m_{31}=m_1,\quad m_{23}-m_{12}=m_2,\quad m_{31}-m_{23}=m_3. \tag{2.40}$$

If $m_1+m_2+m_3=0$ (which is required for consistency of (2.40)) then there is a one parameter family of solutions. Summing (2.39) over this family leads to

$$\frac{\binom{j_2+j_3-j_1}{j_2-j_1-m_3}\binom{j_1+j_2-j_3}{j_2-j_3+m_1}}{(-1)^{m_1-m_3-2j_3-2j_1+j_2}}\, {}_3F_2\left[\begin{array}{ccc} j_2-j_1-j_3, & -j_3-m_3, & -j_1+m_1 \\ 1+j_2-j_3+m_1, & 1+j_2-j_1-m_3 \end{array}\right]. \tag{2.41}$$

The last step is to restore the binomial coefficients from (2.37). This allows the result to be neatly packaged as a 3j-symbol.

$$\langle \mathcal{O}_{j_1,m_1}\mathcal{O}_{j_2,m_2}\mathcal{O}_{j_3,m_3}\rangle = C_{j_1,j_2,j_3}\begin{pmatrix} j_1 & j_2 & j_3 \\ m_1 & m_2 & m_3 \end{pmatrix}\sqrt{\frac{(j_1+j_2-j_3)!(j_2+j_3-j_1)!(j_3+j_1-j_2)!}{(2j_1)!(2j_2)!(2j_3)!(j_1+j_2+j_3+1)!^{-1}}}. \tag{2.42}$$

It is now straightforward to project higher-point functions onto $U(1)_L$ charges by virtue of the OPE.

Considering tree-level four-point functions from [24], our prescription is to find all of the places where $v_{ij}$ factors appear with $C_{j_1,j_2,j_0}C_{j_3,j_4,j_0}$ coefficients and replace them with (2.42). The fact that this yields a product of 3j-symbols is completely obscured if one takes the expressions in (2.25) and repeats the calculation above for all of their terms. It is better to work with each harmonic polynomial as a whole by taking $\langle \mathcal{O}_{j_1}(v_1)\mathcal{O}_{j_2}(v_2)\mathcal{O}_{j_3}(v_3)\mathcal{O}_{j_4}(v_4)\rangle$ and inserting an $SU(2)_L$ covariant projector.

$$\left.\langle \mathcal{O}_{j_1}(v_1)\mathcal{O}_{j_2}(v_2)\mathcal{O}_{j_3}(v_3)\mathcal{O}_{j_4}(v_4)\rangle\right|_{j_0} = \prod_{i<j} v_{ij}^{\gamma_{ij}}(v_{12}v_{34})^{\mathcal{E}-2}\mathcal{Y}_{j_0}(\beta) \tag{2.43}$$

$$= \frac{(\partial_5\cdot\partial_6)^{2j_0}}{(2j_0)!}\langle \mathcal{O}_{j_1}(v_1)\mathcal{O}_{j_2}(v_2)\mathcal{O}_{j_0}(v_5)\rangle\langle \mathcal{O}_{j_0}(v_6)\mathcal{O}_{j_3}(v_3)\mathcal{O}_{j_4}(v_4)\rangle.$$

This allows the main result of this subsection to be derived from four-point functions as well. Note that (2.43) is a special case of the "shadow integral" where this time the measure is discrete because we are working with a compact group [88].

## 3 Lower order input

Our main goal is to compute the spectral density numerators in (2.34). If we had access to the exact double discontinuity of a four-point function with extended supersymmetry, the Lorentzian inversion formula would allow this to be done for all spins. Indeed, the critical spin for a generic CFT was found to be $\ell_* = 2$ in [18]. If (2.31) failed to extract the OPE data for a scalar superconformal primary, this would imply a similar failure for $\ell = 2$ descendants which would be a contradiction. Unfortunately, these are strictly non-perturbative arguments. In perturbation theory, one must analyze the Regge limit to determine $\ell_*$ for each order separately. The present case is similar to [66] in that $\ell_*$ first becomes positive at one loop. It is therefore completely safe at lower orders to use Lorentzian inversion to study all spins.

Section 3.1 begins the process by analyzing the double discontinuity at zeroth order. At first order, there is an opportunity to use the projection formula (2.42) as well. This is pursued in sections 3.2 and 3.3 which explore two different methods for bootstrapping holographic correlators. These are based on the SCFT/chiral algebra correspondence [89] and the Mellin representation [25–27] respectively. The ensuing calculations will make frequent use of the variables

$$x = \frac{z}{1-z}, \qquad y = \frac{1-\bar{z}}{\bar{z}} \tag{3.1}$$

and results will be most conveniently expressed in terms of

$$R_b(h) = r_h \frac{\Gamma(h-b-1)}{\Gamma(h+b+1)}, \tag{3.2}$$

where $r_h$ appeared in (2.31).

### 3.1 Generalized free theory

The most natural starting point is the limit of strictly infinite central charge so that only the disconnected parts of holographic correlators survive. For the Higgs branch $\frac{1}{2}$-BPS operators we have been discussing, it is clear that the dynamical parts of the equal weight four-point function $\left\langle \mathcal{O}_j^{I_1} \mathcal{O}_j^{I_2} \mathcal{O}_j^{I_3} \mathcal{O}_j^{I_4} \right\rangle$ are

$$G^{I_1 I_2 I_3 I_4}(U,V;\alpha,\beta) = \delta^{I_1 I_2}\delta^{I_3 I_4} + (\alpha U)^{2j+2}\beta^{2j}\delta^{I_1 I_3}\delta^{I_2 I_4} + \left[(\alpha-1)\frac{U}{V}\right]^{2j+2}(\beta-1)^{2j}\delta^{I_1 I_4}\delta^{I_2 I_3},$$

$$H^{I_1 I_2 I_3 I_4}(z,\bar{z};\alpha,\beta) = \beta^{2j}\sum_{l=0}^{2j}\alpha^l\sum_{m=0}^{2j-l}z^{l+m+1}\bar{z}^{2j-m+1}\delta^{I_1 I_3}\delta^{I_2 I_4} \tag{3.3}$$

$$+ \frac{(1-\beta)^{2j}}{(z-1)(\bar{z}-1)}\sum_{l=0}^{2j}(1-\alpha)^l\sum_{m=0}^{2j-l}\left(\frac{z}{z-1}\right)^{l+m+1}\left(\frac{\bar{z}}{\bar{z}-1}\right)^{2j-m+1}\delta^{I_1 I_4}\delta^{I_2 I_3},$$

$$f^{I_1 I_2 I_3 I_4}(z;\beta) = \delta^{I_1 I_2}\delta^{I_3 I_4} + z^{2j+2}\beta^{2j}\delta^{I_1 I_3}\delta^{I_2 I_4} + \left(\frac{z}{1-z}\right)^{2j+2}(\beta-1)^{2j}\delta^{I_1 I_4}\delta^{I_2 I_3}.$$

The singlet is the only $SU(2)_L \times SU(2)_R$ representation that all of the auxiliary correlators above contain. We can zoom in on this part with (2.28) or by using the identities in [90].

$$H_a(z,\bar{z};0,0) = \frac{|G_F|(F_u)^0_a}{2j+1}\frac{z\bar{z}}{z-\bar{z}}\sum_{l=0}^{2j}\frac{z^{2j+1}\bar{z}^l - \bar{z}^{2j+1}z^l}{l+1} \tag{3.4}$$

$$- \frac{|G_F|(F_t)^0_a}{2j+1}\frac{z\bar{z}}{z-\bar{z}}\sum_{l=0}^{2j}\frac{1}{l+1}\left[\frac{z^{2j+1}\bar{z}^l}{(z-1)^{2j+2}(\bar{z}-1)^{l+1}} - \frac{\bar{z}^{2j+1}z^l}{(\bar{z}-1)^{2j+2}(z-1)^{l+1}}\right].$$

To handle the flavour part of (3.4), we have recognized that $\delta^{I_1 I_2}\delta^{I_3 I_4} = |G_F| P_0^{I_1 I_2 | I_3 I_4}$. Contracting with a projector in another channel produces an element of the corresponding crossing matrix. These crossing matrices are defined in Appendix B where explicit expressions for the groups in table 1 are given. Only the second term of (3.4) has a double discontinuity as this requires there to be a singularity as $\bar{z} \to 1$. Performing the substitution (3.1) and using the Lorentzian inversion formula, we find

$$
\begin{aligned}
c_a^{(0)}(h,\bar{h};0,0) &= \frac{|G_F|(F_t)^0_a}{2j+1}\frac{r_{\bar{h}}^2}{4\pi^2}\int_0^1\frac{dz}{z^2}k_{1-h}(z)\int_0^1\frac{d\bar{z}}{\bar{z}^2}\frac{k_{\bar{h}}(\bar{z})}{\bar{h}-\frac{1}{2}}\sum_{l=0}^{2j}\mathrm{dDisc}\left[\frac{x^{l+1}y^{-2j-2}-x^{2j+2}y^{-l-1}}{(-1)^{2j+l}(l+1)}\right] \\
&= \frac{|G_F|(F_t)^0_a(-1)^{2j+1}}{(2j+1)(2j+1)!^2}\sum_{l=0}^{2j}\frac{\pi(-1)^l}{l!(l+1)!} \\
&\quad \left[\cot[\pi(2j+2-h)]R_{-2j-2}(h)R_{-l-1}(\bar{h})-\cot[\pi(l+1-h)]R_{-l-1}(h)R_{-2j-2}(\bar{h})\right],
\end{aligned}
\tag{3.5}
$$

after referring to (3.2) and Appendix A. Using both terms, which are needed to examine poles above the double-trace threshold, the residue of $h = 2j + 2 + n$ is

$$
\begin{aligned}
S_a^{(0,0)}(n,\bar{h};0,0) &= 2|G_F|(F_t)^0_a\, r_{2j+2+n}r_{\bar{h}}\,(n+1)_{4j+2}(\bar{h}-2j-1)_{4j+2}(2j+1)!^{-4} \\
&\quad \left[\frac{1}{(2j+n+1)(2j+n+2)}-\frac{1}{\bar{h}(\bar{h}-1)}\right].
\end{aligned}
\tag{3.6}
$$

This proves a formula presented in [44].

We will also need an analogue of (3.6) for $\langle\mathcal{O}_{j,m}\mathcal{O}_{j,-m}\mathcal{O}_{j,m}\mathcal{O}_{j,-m}\rangle$. It is important here that the a double-trace of components is not the same as one component of a double-trace. The double-trace operator $\mathcal{O}_{j,m}\mathcal{O}_{j,-m}$, while $U(1)_L$ neutral of course, receives contributions from the $m_L = 0$ components of $\mathcal{O}_j\mathcal{O}_j$ operators with *all* values of $j_L$. Due to this complication, we will repeat the steps above instead of applying (2.42). This is simple because projecting a disconnected four-point function like this is equivalent to setting $\beta = 0$. The $u$-channel becomes charged and drops out leaving only

$$
G^{I_1 I_2 I_3 I_4}(U,V;\alpha) = \delta^{I_1 I_2}\delta^{I_3 I_4} + \left[(1-\alpha)\frac{U}{V}\right]^{2j+2}\delta^{I_1 I_4}\delta^{I_2 I_3}.
\tag{3.7}
$$

The double discontinuity of the auxiliary correlator therefore has the same form as before. It is just larger by a factor of $2j + 1$ due to there being no $SU(2)_L$ inner product to take. Due to the lack of a $u$-channel in (3.7) it is also clear that the second term in the Lorentzian inversion formula must vanish. In other words, (2.28) no longer holds because the first two operators transform differently under $U(1)_L$. This leads to OPE coefficients that are larger by a factor of $j + \frac{1}{2}$. The exception occurs when $m = 0$. All in all,

$$
S_a^{(0,0)}(n,\bar{h};0) = (1+\delta_{m,0})\left(j+\frac{1}{2}\right)S_a^{(0,0)}(n,\bar{h};0,0).
\tag{3.8}
$$

This relation also could have been guessed from the three-point function

$$
\langle\mathcal{O}_{j_1}(v_1)\mathcal{O}_{j_2}(v_2)[\mathcal{O}_{j_1}\mathcal{O}_{j_2}]_{j_3}(v_3)\rangle = \sqrt{\frac{(2j_1)!(2j_2)!(2j_3+1)(1+(-1)^{j_1+j_2-j_3}\delta_{j_1,j_2})}{(j_1+j_2-j_3)!(j_1+j_2+j_3+1)!}}\\
v_{12}^{j_1+j_2-j_3}v_{23}^{j_2+j_3-j_1}v_{31}^{j_3+j_1-j_2},
\tag{3.9}
$$

which holds for pure $SU(2)_L$ representations.[18]

---

[18] The derivation of (3.9) in [90] refers to the R-symmetry as $SO(8)$ because the generalized free theory discussed here also appears in the topological subsector of 3d $\mathcal{N} = 8$ SCFTs [83].

## 3.2  Tree-level data from the chiral algebra

We have previously referred to the last line of (3.3) as "the chiral correlator" for good reason. It is a four-point function of conserved currents in a 2d CFT whose OPEs descend from four dimensions as discovered in [89]. These currents come from so called *Schur operators* which live in short multiplets and exist in any 4d $\mathcal{N} = 2$ SCFT. Under a particular nilpotent supercharge, their cohomology classes cease to depend on $\bar{z}$ as they are translated away from the origin along the locus suggested by (2.19). A distinguishing feature of the holographic theories studied here is that they admit a subsector of single-trace operators which are all $\frac{1}{2}$-BPS. As shown in [24], this is enough to guarantee that the chiral algebra they generate is unique at order $1/c_J$. Moreover, higher powers of $1/c_J$ that modify a given correlator truncate at a finite order which is linear in the external weights [90]. The simplest case is that of $\langle \mathcal{O}_0 \mathcal{O}_0 \mathcal{O}_0 \mathcal{O}_0 \rangle$ which yields the *exact* function

$$f^{I_1 I_2 I_3 I_4}(z) = \delta^{I_1 I_2} \delta^{I_3 I_4} + z^2 \delta^{I_1 I_3} \delta^{I_2 I_4} + \left( \frac{z}{1-z} \right)^2 \delta^{I_1 I_4} \delta^{I_2 I_3} \tag{3.10}$$
$$+ \frac{6}{c_J} \left[ z f^{I_1 I_3 J} f^{J I_2 I_4} + \frac{z}{z-1} f^{I_1 I_4 J} f^{J I_2 I_3} \right].$$

The loop-level data to be calculated later will therefore only affect $H^{I_1 I_2 I_3 I_4}(z, \bar{z})$. For now, we will demonstrate an interesting technique which uses interplay between these two correlators to compute anomalous dimensions at tree-level. The point is that the crossing equation

$$H^{I_1 I_2 I_3 I_4}(z, \bar{z}) = \left( \frac{z}{z-1} \frac{\bar{z}}{\bar{z}-1} \right)^2 H^{I_3 I_2 I_1 I_4}(1-z, 1-\bar{z}) \tag{3.11}$$
$$- \frac{z(1-z)^{-1} f^{I_1 I_2 I_3 I_4}(\bar{z}) - \bar{z}(1-\bar{z})^{-1} f^{I_1 I_2 I_3 I_4}(z)}{z - \bar{z}}$$

is inherited from (3.10). One can therefore find a simple characterization of all poles as $\bar{z} \to 1$, *i.e.* all tree-level contributions to the double discontinuity. They are given by the terms in $H^{I_3 I_2 I_1 I_4}(V, U)$ with at most one power of $V$. The functions of $U$ which accompany them are known as lightcone blocks [91]. The associated operators are necessarily in short multiplets since operators above the double-trace threshold start at $V^2$. As a result, their OPE coefficients are protected and contained in (3.10). Consider the expansion

$$f^{I_3 I_2 I_1 I_4}(z) = \delta^{I_2 I_3} \delta^{I_1 I_4} + \sum_{l=0}^{\infty} \frac{(l+1)(l+1)!}{(l+3)_l} \left[ \delta^{I_3 I_4} \delta^{I_1 I_2} + (-1)^l \delta^{I_1 I_3} \delta^{I_2 I_4} \right] k_{l+2}(z) \tag{3.12}$$
$$+ \frac{6}{c_J} \sum_{l=0}^{\infty} \frac{l!^2}{(2l)!} \left[ f^{I_3 I_4 J} f^{J I_1 I_2} - (-1)^l f^{I_1 I_3 J} f^{J I_2 I_4} \right] k_{l+1}(z).$$

Table 4 shows that the sum of 4d conformal blocks corresponding to this is

$$\sum_{\ell=0}^{\infty} \frac{(\ell+1)(\ell+1)!}{(\ell+3)_\ell} \left[ \delta^{I_3 I_4} \delta^{I_1 I_2} + (-1)^\ell \delta^{I_1 I_3} \delta^{I_2 I_4} \right] V^{-1} g_{\ell+4, \ell}(V, U) \tag{3.13}$$
$$+ \frac{6}{c_J} \sum_{\ell=0}^{\infty} \frac{(\ell+1)!^2}{(2\ell+2)!} \left[ f^{I_3 I_4 J} f^{J I_1 I_2} + (-1)^\ell f^{I_1 I_3 J} f^{J I_2 I_4} \right] V^{-1} g_{\ell+4, \ell}(V, U).$$

Taking the leading lightcone limit everywhere, [91] instructs us to write (3.13) as

$$\frac{V}{(1-U)^2}\sum_{\ell=0}^{\infty}\frac{(\ell+1)(\ell+1)!}{(\ell+3)_\ell}\left[\delta^{I_3I_4}\delta^{I_1I_2}+(-1)^\ell\delta^{I_1I_3}\delta^{I_2I_4}\right]k_{\ell+2}(1-U) \tag{3.14}$$

$$+\frac{6}{c_J}\frac{V}{(1-U)^2}\sum_{\ell=0}^{\infty}\frac{(\ell+1)!^2}{(2\ell+2)!}\left[f^{I_3I_4J}f^{JI_1I_2}+(-1)^\ell f^{I_1I_3J}f^{JI_2I_4}\right]k_{\ell+2}(1-U)$$

$$=V\delta^{I_1I_3}\delta^{I_2I_4}+\frac{V}{U^2}\delta^{I_3I_4}\delta^{I_1I_2}-\frac{6}{c_J}\left[\frac{V}{1-U}f^{I_1I_3J}f^{JI_2I_4}-\frac{V}{U(1-U)}f^{I_3I_4J}f^{JI_1I_2}\right]$$

$$+\frac{6}{c_J}\frac{V}{(1-U)^2}\left[f^{I_1I_3J}f^{JI_2I_4}-f^{I_3I_4J}f^{JI_1I_2}\right]k_1(1-U).$$

The first line has been resummed into power laws since it retains the exact same form that it had in (3.12). The second line has become more interesting due to the single-trace $k_1(z)$ block in the chiral correlator which does not have a 4d counterpart. After using the Jacobi identity

$$f^{I_1I_2J}f^{JI_3I_4}+f^{I_1I_4J}f^{JI_2I_3}+f^{I_1I_3J}f^{JI_4I_2}=0, \tag{3.15}$$

it turns out to be proportional to a single flavour structure. Since (3.14) now isolates the desired factor of $V$, it is ready to be plugged into (3.11). In the disconnected part, all of the terms with $\delta^{I_1I_2}\delta^{I_3I_4}$ and $\delta^{I_1I_4}\delta^{I_2I_3}$ either cancel or become regular so that the double discontinuity is precisely the one we saw in (3.5). The $1/c_J$ calculation reveals a similar pattern and produces

$$\text{dDisc}\left[(z-\bar{z})H^{I_1I_2I_3I_4}(z,\bar{z})\right]=\text{dDisc}\left[\left(\frac{x^2}{y}-\frac{x}{y^2}\right)\delta^{I_1I_4}\delta^{I_2I_3}+\frac{6}{c_J}\frac{x}{y}f^{I_1I_4J}f^{JI_2I_3}\right] \tag{3.16}$$

$$+\text{dDisc}\left[\frac{6}{c_J}\frac{x^2}{y}\log\left(\frac{x}{x+1}\right)f^{I_1I_4J}f^{JI_2I_3}\right].$$

This exhibits the telltale sign of anomalous dimensions.

## 3.3 Tree-level data from Mellin space

The lesson above is that the superconformal Ward identity is enough to determine the tree-level four-point function as long as all long multiplets are double-trace. We only established this for $\langle\mathcal{O}_0\mathcal{O}_0\mathcal{O}_0\mathcal{O}_0\rangle$ but the steps needed to extend the derivation to mixed correlators are purely technical [86]. Thanks to an algorithm developed in [11], it is also possible to impose the superconformal Ward identity in Mellin space where the ansatz can be organized in a more intuitive way. This Mellin space approach is what [24] used to solve for all higher weight cousins of the 4d correlator (1.3) (along with analogous amplitudes in other dimensions where there is no alternative).

### 3.3.1 Full and auxiliary Mellin amplitudes

To explain the key features, conformal cross ratios in (2.14) are traded for Mandelstam variables which satisfy

$$s+t+u=\sum_{i=1}^{4}\Delta_i\equiv\sum_{i=1}^{4}(2j_i+2). \tag{3.17}$$

To each single-trace supermultiplet we associate an object

$$\mathcal{S}_j(s,t;\alpha)=\sum_{(\Delta,\ell,j_R)\in B\bar{B}[0;0]_{2j+2}^{(2j+2,0)}}\lambda_{\Delta,\ell,j_R}\mathcal{Y}_{j_R}(\alpha)\mathcal{M}_{\Delta,\ell}(s,t), \tag{3.18}$$

whose coefficients are exactly those which appear in the corresponding superconformal block. Each function of $s$ and $t$ is the polar part of an ordinary conformal block which is the same as the polar part of an exchange Witten diagram [92]. A proper holographic ansatz includes (3.18) summed over $\mathcal{I}_{12|34} \equiv \{\min(|j_{12}|, |j_{34}|) + 1, \ldots, \max(j_1 + j_2, j_3 + j_4) - 1\}$ in all three channels.[19] It should also involve a contact term in the form of a crossing symmetric (including Bose symmetric) polynomial whose degree is one less than the maximal spin that appears in (3.18). Explicitly,

$$\mathcal{M}^{I_1 I_2 I_3 I_4}(s,t;\alpha,\beta) = f^{I_1 I_3 J} f^{J I_2 I_4} \sum_{j \in \mathcal{I}_{13|24}} C_{j_1,j_3,j} C_{j_2,j_4,j} \alpha^{\mathcal{E}} \beta^{\mathcal{E}-2} \mathcal{Y}_j(\tfrac{1}{\beta}) \mathcal{S}_j(u,t;\tfrac{1}{\alpha}) \qquad (3.19)$$

$$+ f^{I_1 I_4 J} f^{J I_3 I_2} \sum_{j \in \mathcal{I}_{14|32}} C_{j_1,j_4,j} C_{j_3,j_2,j} (\alpha-1)^{\mathcal{E}} (\beta-1)^{\mathcal{E}-2} \mathcal{Y}_j\left(\tfrac{\beta}{\beta-1}\right) \mathcal{S}_j\left(t,s;\tfrac{\alpha}{\alpha-1}\right)$$

$$+ f^{I_1 I_2 J} f^{J I_3 I_4} \sum_{j \in \mathcal{I}_{12|34}} C_{j_1,j_2,j} C_{j_3,j_4,j} \mathcal{Y}_j(\beta) \mathcal{S}_j(s,t;\alpha) + \mathcal{C}^{I_1 I_2 I_3 I_4}(\alpha,\beta).$$

The superconformal Ward identity can then be used to solve for $C_{j_1,j_2,j_3}$ and $\mathcal{C}^{I_1 I_2 I_3 I_4}(\alpha,\beta)$ at low-lying weights until a pattern becomes apparent.[20] This is greatly facilitated by exploiting the freedom to have the residue of a pole in one Mandelstam variable formally depend on *both* of the others. In particular, the contact term comes out to zero if we use a remarkable prescription in [19, 20] for symmetrizing at the level of supermultiplets.[21]

The relation between a (flavour projected) position space four-point function and its Mellin amplitude is

$$G_a(U,V;\alpha,\beta) = \int_{-i\infty}^{i\infty} \frac{\mathrm{d}s\mathrm{d}t}{(4\pi i)^2} U^{\frac{s}{2}-j_1-j_2-2+\mathcal{E}} V^{\frac{t}{2}+j_1-j_4-\mathcal{E}} \mathcal{M}_a(s,t;\alpha,\beta) \Gamma_{\{\Delta_i\}}(s,t) \qquad (3.20)$$

$$\Gamma_{\{\Delta_i\}}(s,t) \equiv \Gamma\left[\tfrac{\Delta_1+\Delta_2-s}{2}\right] \Gamma\left[\tfrac{\Delta_3+\Delta_4-s}{2}\right] \Gamma\left[\tfrac{\Delta_1+\Delta_4-t}{2}\right] \Gamma\left[\tfrac{\Delta_2+\Delta_3-t}{2}\right] \Gamma\left[\tfrac{\Delta_1+\Delta_3-u}{2}\right] \Gamma\left[\tfrac{\Delta_2+\Delta_4-u}{2}\right],$$

with the contour chosen to encircle poles in $s$ and $t$ but not $u$. Powers of $U$ and $V$ which multiply the correlator can therefore be translated into difference operators.

$$\widehat{U^m V^n} \circ \mathcal{M}(s,t) = \frac{\Gamma_{\{\Delta_i\}}(s-2m,t-2n)}{\Gamma_{\{\Delta_i\}}(s,t)} \mathcal{M}(s-2m,t-2n). \qquad (3.21)$$

The Mellin superconformal Ward identity is formulated by using (3.21) in the sum and difference of the $z$ and $\bar{z}$ versions of (2.19) [11]. There is also an auxiliary Mellin amplitude for 4d $\mathcal{N} = 2$ theories defined by

$$H_a(U,V;\alpha,\beta) = \int_{-i\infty}^{i\infty} \frac{\mathrm{d}s\mathrm{d}t}{(4\pi i)^2} U^{\frac{s}{2}-j_1-j_2-2+\mathcal{E}} V^{\frac{t}{2}+j_1-j_4-\mathcal{E}} \widetilde{\mathcal{M}}_a(s,t;\alpha,\beta) \widetilde{\Gamma}_{\{\Delta_i\}}(s,t) \qquad (3.22)$$

$$\widetilde{\Gamma}_{\{\Delta_i\}}(s,t) \equiv \Gamma\left[\tfrac{\Delta_1+\Delta_2-s}{2}\right] \Gamma\left[\tfrac{\Delta_3+\Delta_4-s}{2}\right] \Gamma\left[\tfrac{\Delta_1+\Delta_4-t}{2}\right] \Gamma\left[\tfrac{\Delta_2+\Delta_3-t}{2}\right] \Gamma\left[\tfrac{\Delta_1+\Delta_3-\tilde{u}}{2}\right] \Gamma\left[\tfrac{\Delta_2+\Delta_4-\tilde{u}}{2}\right],$$

where $\tilde{u} \equiv u - 2$ so that crossing still acts by a permutation.[22] In terms of a new difference

---

[19]The upper and lower limits are moved in by one compared to selection rules since extremal correlators vanish in AdS/CFT. This is a consequence of single-particle and double-particle states being orthogonal [69, 70].

[20]As explained in [24], this pattern can also be obtained by studying the flat space limit of (3.19). Actually proving that it holds for all weights is difficult in general but a method that works in four dimensions was discussed in [90].

[21]Another technique proposed in [19, 20] is helpful for determining the coefficients in (3.18) when they are not already known.

[22]Another common name for (3.22) is the reduced amplitude. This should not be confused with the reduced amplitude of [25] which is $\mathcal{M}(s,t) \Gamma_{\{\Delta_i\}}(s,t)$.

operator defined by

$$\widehat{\widetilde{U^m V^n}} \circ \mathcal{M}(s,t) = \frac{\widetilde{\Gamma}_{\{\Delta_i\}}(s-2m, t-2n)}{\Gamma_{\{\Delta_i\}}(s,t)} \widetilde{\mathcal{M}}(s-2m, t-2n), \tag{3.23}$$

the translation of (2.20) into Mellin space is

$$\mathcal{M}_a(s,t;\alpha,\beta) = \left[1 - \alpha(1 + \widehat{\widetilde{U}} - \widehat{\widetilde{V}}) + \alpha^2 \widehat{\widetilde{U}}\right] \circ \widetilde{\mathcal{M}}_a(s,t;\alpha,\beta). \tag{3.24}$$

The chiral correlator, which appears to be missing, is in fact part of the auxiliary Mellin amplitude due to the contour pinching mechanism described in [10].

### 3.3.2 Spectral density at tree-level

With this formalism in place, we can return to the goal of setting up a loop calculation. For $\langle \mathcal{O}_0 \mathcal{O}_0 \mathcal{O}_j \mathcal{O}_j \rangle$, the auxiliary Mellin amplitude is

$$\widetilde{\mathcal{M}}^{I_1 I_2 I_3 I_4}(s,t) = -\frac{24}{c_J(2j)!} \left[ \frac{f^{I_1 I_2 J} f^{J I_3 I_4}}{(s-2)(\tilde{u}-2j-2)} - \frac{f^{I_1 I_4 J} f^{J I_2 I_3}}{(t-2j-2)(\tilde{u}-2j-2)} \right]. \tag{3.25}$$

For $\langle \mathcal{O}_{0,0} \mathcal{O}_{0,0} \mathcal{O}_{j,m} \mathcal{O}_{j,-m} \rangle$, it is the same because these weights yield the simplest possible 3j-symbol in (2.42). While there are various methods for decomposing (3.25) in a given channel [93], the double-trace data should be extracted in a form which is manifestly analytic in spin. This can be done by isolating the pole at $t = 2j+2$ in order to solve for the double discontinuity. Using (3.22),

$$H^{I_1 I_2 I_3 I_4}(U,V) = -\frac{6}{c_J} \frac{V^{-1}}{(2j)!} f^{I_1 I_4 J} f^{J I_2 I_3} \int_{-i\infty}^{i\infty} \frac{ds}{2\pi i} \frac{U^{\frac{s}{2}}}{2-s} \Gamma\left(2-\frac{s}{2}\right) \Gamma\left(2j+2-\frac{s}{2}\right) \Gamma\left(\frac{s}{2}\right)^2 + \dots$$

$$= -\frac{3}{c_J} \frac{V^{-1}}{(2j)!} f^{I_1 I_4 J} f^{J I_2 I_3} \int_{-i\infty}^{i\infty} \frac{ds}{2\pi i} U^{\frac{s}{2}} \Gamma\left(1-\frac{s}{2}\right) \Gamma\left(2j+2-\frac{s}{2}\right) \Gamma\left(\frac{s}{2}\right)^2 + \dots$$

$$= -\frac{6}{c_J} \frac{2j+1}{2j+2} V^{-1}{}_2F_1\left(1, 2j+2; 2j+3; 1-U^{-1}\right) f^{I_1 I_4 J} f^{J I_2 I_3} + \dots, \tag{3.26}$$

where the missing terms are higher order in $V$.[23] The $j = 0$ case precisely reproduces (3.16). It is now possible to compute the spectral density with Lorentzian inversion. Making the replacement $f^{I_1 I_2 J} f^{J I_3 I_4} = G_F^\vee P_1^{I_1 I_2 | I_3 I_4}$, the integral evaluates to

$$c_a^{(1)}(h, \bar{h}) = 6 G_F^\vee (F_t)^1{}_a (2j+1) \frac{r_{\bar{h}}^2}{4\pi^2} \int_0^1 \frac{dz}{z^2} k_{1-h}(z) \int_0^1 \frac{d\bar{z}}{\bar{z}^2} \frac{k_{\bar{h}}(\bar{z})}{\bar{h}-\frac{1}{2}} \tag{3.27}$$

$$\text{dDisc}\left[ \frac{(-x)^{2j+2}}{y} \log\left(\frac{x}{x+1}\right) + \sum_{l=0}^{2j} \frac{(-1)^l}{l-2j-1} \frac{x^{l+1}}{y} \right]$$

$$= -6 G_F^\vee (F_t)^1{}_a (2j+1) \left[ (-1)^{2j} \left( \pi^2 \csc^2[\pi(2j+2-h)] - \pi \cot[\pi(2j+2-h)] \frac{\partial}{\partial q} \right) \right.$$

$$\left. \frac{R_1(\bar{h}) R_{-q}(h)}{\Gamma(q)^2} \bigg|_{q=2j+2} - \sum_{l \neq 2j+1} \frac{(-1)^l}{l-2j-1} \frac{\pi \cot[\pi(l+1-h)]}{l!^2} R_1(\bar{h}) R_{-l-1}(h) \right].$$

---

[23]Since it comes from the difference operator (3.23), the single-trace $\tilde{u}$ pole lies to the right of the contour in contrast to the original $u$ poles which lie to the left. This is what allowed us to absorb $\frac{1}{2-s}$ into the gamma function on the second line.

The behaviour near $h = n + 2$ is governed by

$$S_a^{(1,1)}(n,\bar{h}) = 24(-1)^{2j} G_F^\vee (F_t)^1_a\, r_{\bar{h}} r_{n+2} (n - 2j + 1)_{4j+2} (2j)!^{-1} (2j+1)!^{-1}\,, \qquad (3.28)$$

$$S_a^{(1,0)}(n,\bar{h}) = 12(-1)^{2j} G_F^\vee (F_t)^1_a\, \frac{\partial}{\partial h} R_{-1}(\bar{h}) R_{-2j-2}(h)\bigg|_{h=n+2} (2j)!^{-1} (2j+1)!^{-1}\,,$$

where the first line is immediate and the second requires the identities of [66].

# 4 Going to one loop

Our results thus far can be summarized by the following averaged quantities where the subscripts denote flavour representation, twist family and spin respectively. At zeroth order,

$$\left\langle a^{(0)}\right\rangle_{a,n,\ell} = \frac{2|G_F|(F_t)^0_a (n+1)_{4j+2}(n+\ell+2)_{4j+2}\, r_{2j+2+n} r_{2j+3+n+\ell}}{(2j+1)!^4}$$
$$\left[\frac{1}{(2j+n+1)(2j+n+2)} - \frac{1}{(2j+n+\ell+2)(2j+n+\ell+3)}\right]\,, \qquad (4.1)$$

for $SU(2)_L \times SU(2)_R$ singlets in $\left\langle \mathcal{O}_j \mathcal{O}_j \mathcal{O}_j \mathcal{O}_j \right\rangle$ and

$$\left\langle a^{(0)}\right\rangle_{a,n,\ell} = \left(1 + \delta_{m,0}\right)\left(j + \frac{1}{2}\right) \frac{2|G_F|(F_t)^0_a (n+1)_{4j+2}(n+\ell+2)_{4j+2}\, r_{2j+2+n} r_{2j+3+n+\ell}}{(2j+1)!^4}$$
$$\left[\frac{1}{(2j+n+1)(2j+n+2)} - \frac{1}{(2j+n+\ell+2)(2j+n+\ell+3)}\right]\,, \qquad (4.2)$$

for $U(1)_L \times SU(2)_R$ singlets in $\left\langle \mathcal{O}_{j,m} \mathcal{O}_{j,-m} \mathcal{O}_{j,m} \mathcal{O}_{j,-m}\right\rangle$. At first order,

$$\left\langle a^{(0)}\gamma^{(1)}\right\rangle_{a,n,\ell} = (-1)^{2j} \frac{24 G_F^\vee (F_t)^1_a}{(2j)!(2j+1)!} R_{-1}(\bar{h}) R_{-2j-2}(h)\bigg|_{\substack{h=n+2 \\ \bar{h}=n+3+\ell}}$$

$$\left\langle a^{(1)}\right\rangle_{a,n,\ell} = (-1)^{2j} \frac{12 G_F^\vee (F_t)^1_a}{(2j)!(2j+1)!} \left[\frac{\partial}{\partial h} + \frac{\partial}{\partial \bar{h}}\right] R_{-1}(\bar{h}) R_{-2j-2}(h)\bigg|_{\substack{h=n+2 \\ \bar{h}=n+3+\ell}} \qquad (4.3)$$

describe both $\left\langle \mathcal{O}_0 \mathcal{O}_0 \mathcal{O}_j \mathcal{O}_j \right\rangle$ and $\left\langle \mathcal{O}_{0,0} \mathcal{O}_{0,0} \mathcal{O}_{j,m} \mathcal{O}_{j,-m}\right\rangle$ which necessarily exchange only singlets in the auxiliary correlator.

We will mostly not need the second line of (4.3) but the other data will be fed into the machinery of a loop calculation. This will introduce the first non-trivial dependence on the S-fold parameter $k$. The details are explained in section 4.1 where a new average $\left\langle a^{(0)}\gamma^{(1)2}\right\rangle_{a,n,\ell}$ is computed. Various manipulations are required before it can be used to extract anomalous dimensions at one-loop. Section 4.2 discusses these and uses insights from [17] to develop a method which works for the CFTs of [35] and would also work for many others. This equips us to solve for all $\left\langle a^{(0)}\gamma^{(2)}\right\rangle_{a,n,\ell}$ to arbitrarily many orders in $1/\ell$. Section 4.3 presents the results and also (less rigorously) solves for a subset of anomalous dimensions at finite $\ell$. Since there are two central charges in theories with flavour, loop diagrams at order $1/c_J^2$ need to be combined with tree diagrams at order $1/c_T$, a computation which is done in section 4.4. Finally, section 4.5 verifies many of our results using Mellin space. This culminates in the solution for the $k = 2$ one-loop Mellin amplitude which may be of independent interest.

## 4.1 The AdS unitarity method

Much of the progress in AdS loop calculations rests on the fact that a key one-loop term in the spectral density (2.33) depends only on tree-level data. We can see this more explicitly by perturbing scaling dimensions and OPE coefficients away from their $c_J \to \infty$ double-trace values in the conformal block expansion.

$$\left(a_{n,\ell}^{(0)} + c_J^{-1} a_{n,\ell}^{(1)} + c_J^{-2} a_{n,\ell}^{(2)} + \dots\right) g_{\Delta_{n,\ell} + c_J^{-1}\gamma_{n,\ell}^{(1)} + c_J^{-2}\gamma_{n,\ell}^{(2)} + \dots, \ell}(U,V) = a_{n,\ell}^{(0)} g_{\Delta_{n,\ell},\ell}(U,V) \tag{4.4}$$

$$+ c_J^{-1}\left[a_{n,\ell}^{(1)} + a_{n,\ell}^{(0)}\gamma_{n,\ell}^{(1)}\frac{\partial}{\partial\Delta}\right] g_{\Delta_{n,\ell},\ell}(U,V) + c_J^{-2}\left[a_{n,\ell}^{(2)} + \left(a_{n,\ell}^{(1)}\gamma_{n,\ell}^{(1)} + a_{n,\ell}^{(0)}\gamma_{n,\ell}^{(2)}\right)\frac{\partial}{\partial\Delta}\right] g_{\Delta_{n,\ell},\ell}(U,V)$$

$$+ \frac{1}{2}c_J^{-2} a_{n,\ell}^{(0)}\gamma_{n,\ell}^{(1)2}\frac{\partial^2}{\partial\Delta^2} g_{\Delta_{n,\ell},\ell}(U,V).$$

The middle line of (4.4) generates $\log U$ terms which are ubiquitous. Single logarithms have a vanishing double discontinuity and they are produced every time one expands a direct channel block in the lightcone limit of a crossed channel [14, 15]. The $\log^2 U$ from the last line however is much more rigid. Knowing that

$$\mathrm{dDisc}\left[\log^2(1-\bar{z})\right] = 4\pi^2, \tag{4.5}$$

such a term plays an essential role in the crossing equation (3.11). It is the contribution to the double discontinuity which is intrinsic to infinite sums and cannot be seen by looking for individual exchanges below the double-trace threshold. After applying (2.31), it becomes clear that CFTs without degeneracy have their loop-level data determined by $a_{n,\ell}^{(0)}\gamma_{n,\ell}^{(1)2}$ at tree-level in the same correlator. This was first shown in [45], slightly before the discovery of the Lorentzian inversion formula, and phrased in terms of an equivalent formulation of the analytic bootstrap [16].

Our present task is to apply this AdS unitarity method in the presence of degeneracy which requires multiple correlators. A single one would not be enough since

$$\left\langle a^{(0)}\gamma^{(1)2}\right\rangle_{n,\ell} \neq \left\langle a^{(0)}\gamma^{(1)}\right\rangle_{n,\ell}^2 / \left\langle a^{(0)}\right\rangle_{n,\ell}. \tag{4.6}$$

Consider the $\mathcal{O}_0 \times \mathcal{O}_0$ OPE in the $(0, 0, \mathbf{R}_a)$ representation of $SU(2)_L \times SU(2)_R \times G_F$ with some spin $\ell$ of the appropriate parity. By definition, only $[\mathcal{O}_0\mathcal{O}_0]_n$ double-traces contribute at leading order. It is also true, at least generically, that each one is a non-trivial linear combination of some $(\Phi_{\underline{1}}, \Phi_{\underline{2}}, \dots)$ which diagonalize dilations on the $\Delta = 4 + 2n + \ell$ subspace at the next order. We can invert this to say that operators which have definite scaling dimensions, and can therefore help us learn about $\langle\mathcal{O}_0\mathcal{O}_0\mathcal{O}_0\mathcal{O}_0\rangle$, contain information about double-traces in other four-point functions. The bulk interpretation of this statement is that there are many intermediate states which can run in the loop. Proceeding as in [63, 64], double-traces that mix for a given $n$ are

$$[\mathcal{O}_0\mathcal{O}_0]_n, [\mathcal{O}_1\mathcal{O}_1]_{n-2}, \dots, [\mathcal{O}_{\lfloor n/2 \rfloor}\mathcal{O}_{\lfloor n/2 \rfloor}]_{n \bmod 2}. \tag{4.7}$$

In the original 7-brane backgrounds of [37], $n$ would be lowered by both even and odd integers. We have restricted to even integers here because only these survive a $k = 2$ S-fold. A total of $\lfloor\frac{n}{2}\rfloor + 1$ operators from the new basis therefore enter into (4.1) and (4.3) according to

$$\left\langle a^{(0)}\right\rangle_{n-2j} = \lambda_{j,j,\underline{1}}^2 + \dots + \lambda_{j,j,\underline{\lfloor n/2 \rfloor+1}}^2, \tag{4.8}$$

$$\left\langle a^{(0)}\gamma^{(1)}\right\rangle_n = \lambda_{0,0,\underline{1}}\lambda_{j,j,\underline{1}}\gamma_{\underline{1}} + \dots + \lambda_{0,0,\underline{\lfloor n/2 \rfloor+1}}\lambda_{j,j,\underline{\lfloor n/2 \rfloor+1}}\gamma_{\underline{\lfloor n/2 \rfloor+1}}.$$

Adding a $(j_1, j_2)$ superscript to indicate a correlator of $\left\langle \mathcal{O}_{j_1} \mathcal{O}_{j_1} \mathcal{O}_{j_2} \mathcal{O}_{j_2} \right\rangle$, the first line of (4.8) shows that the change of basis matrix is given by

$$
\begin{pmatrix} [\mathcal{O}_0 \mathcal{O}_0]_n \\ \vdots \\ [\mathcal{O}_{\lfloor n/2 \rfloor} \mathcal{O}_{\lfloor n/2 \rfloor}]_{n \bmod 2} \end{pmatrix} = Q \begin{pmatrix} \Phi_{\underline{1}} \\ \vdots \\ \Phi_{\underline{\lfloor n/2 \rfloor + 1}} \end{pmatrix} \tag{4.9}
$$

$$
\equiv \begin{pmatrix} \dfrac{\lambda_{0,0,\underline{1}}}{\sqrt{\left\langle a^{(0)} \right\rangle_n^{(0,0)}}} & \cdots & \dfrac{\lambda_{0,0,\underline{\lfloor n/2 \rfloor + 1}}}{\sqrt{\left\langle a^{(0)} \right\rangle_n^{(0,0)}}} \\ \vdots & \ddots & \vdots \\ \dfrac{\lambda_{\lfloor n/2 \rfloor, \lfloor n/2 \rfloor, \underline{1}}}{\sqrt{\left\langle a^{(0)} \right\rangle_{n \bmod 2}^{(\lfloor n/2 \rfloor, \lfloor n/2 \rfloor)}}} & \cdots & \dfrac{\lambda_{\lfloor n/2 \rfloor, \lfloor n/2 \rfloor, \underline{\lfloor n/2 \rfloor + 1}}}{\sqrt{\left\langle a^{(0)} \right\rangle_{n \bmod 2}^{(\lfloor n/2 \rfloor, \lfloor n/2 \rfloor)}}} \end{pmatrix} \begin{pmatrix} \Phi_{\underline{1}} \\ \vdots \\ \Phi_{\underline{\lfloor n/2 \rfloor + 1}} \end{pmatrix}.
$$

The fact that no double-traces are shared between $\mathcal{O}_{j_1} \times \mathcal{O}_{j_1}$ and $\mathcal{O}_{j_2} \times \mathcal{O}_{j_2}$ ensures that $Q$ is orthogonal. This means that the symmetric matrix $M \equiv Q \operatorname{diag}(\gamma_{\underline{1}}, \dots, \gamma_{\underline{\lfloor n/2 \rfloor + 1}}) Q^{\mathsf{T}}$ satisfies

$$
M^l = \begin{pmatrix} \dfrac{\left\langle a^{(0)} \gamma^{(1)l} \right\rangle_n^{(0,0)}}{\left\langle a^{(0)} \right\rangle_n^{(0,0)}} & \cdots & \dfrac{\left\langle a^{(0)} \gamma^{(1)l} \right\rangle_n^{(0,\lfloor n/2 \rfloor)}}{\sqrt{\left\langle a^{(0)} \right\rangle_n^{(0,0)} \left\langle a^{(0)} \right\rangle_{n \bmod 2}^{(\lfloor n/2 \rfloor, \lfloor n/2 \rfloor)}}} \\ \vdots & \ddots & \vdots \\ \dfrac{\left\langle a^{(0)} \gamma^{(1)l} \right\rangle_n^{(\lfloor n/2 \rfloor, 0)}}{\sqrt{\left\langle a^{(0)} \right\rangle_n^{(0,0)} \left\langle a^{(0)} \right\rangle_{n \bmod 2}^{(\lfloor n/2 \rfloor, \lfloor n/2 \rfloor)}}} & \cdots & \dfrac{\left\langle a^{(0)} \gamma^{(1)l} \right\rangle_{n \bmod 2}^{(\lfloor n/2 \rfloor, \lfloor n/2 \rfloor)}}{\left\langle a^{(0)} \right\rangle_{n \bmod 2}^{(\lfloor n/2 \rfloor, \lfloor n/2 \rfloor)}} \end{pmatrix}. \tag{4.10}
$$

Clearly, (4.1) and (4.3) are the averages which appear in the first row and first column of $M$. The upper left corner of $M^2$, which must be the desired $\left\langle a^{(0)} \gamma^{(1)2} \right\rangle_n^{(0,0)} / \left\langle a^{(0)} \right\rangle_n^{(0,0)}$, is therefore a square of the known tree-level data. Reverting to the previous notation (where $j$ is implicit),

$$
\left\langle a^{(0)} \gamma^{(1)2} \right\rangle_{a,n,\ell} = \sum_{j=0}^{\lfloor \frac{n}{2} \rfloor} \frac{\left\langle a^{(0)} \gamma^{(1)} \right\rangle_{a,n,\ell}^2}{\left\langle a^{(0)} \right\rangle_{a,n-2j,\ell}} \qquad (k = 2) \tag{4.11}
$$

holds for $k = 2$ S-folds.

In all other cases, basis elements of (4.7) must be replaced by double-traces of $U(1)_L$ components. The resulting increase in the dimensionality of $Q$ (where we recall that $[\mathcal{O}_{j,m} \mathcal{O}_{j,-m}]_{n,\ell} = [\mathcal{O}_{j,-m} \mathcal{O}_{j,m}]_{n,\ell}$) depends on the value of $k$. Before implementing the $k | 2m_L$ condition, it is helpful to consider $k = 2$ or even $k = 1$ (no S-fold) and check that they can be handled in $U(1)_L$ language where $SU(2)_L$ symmetry is ignored. Describing $k = 1$, the analogue of (4.11) is found by letting $j$ run over $\mathbb{Z}/2$.

$$
\left\langle a^{(0)} \gamma^{(1)2} \right\rangle_{a,n,\ell} = \sum_{2j=0}^{n} \frac{\left\langle a^{(0)} \gamma^{(1)} \right\rangle_{a,n,\ell}^2}{\left\langle a^{(0)} \right\rangle_{a,n-2j,\ell}} \qquad (k = 1). \tag{4.12}
$$

The relation between (4.1) and (4.2) makes the check succeed. When $j$ is a half-integer, each numerator splits into $j + \frac{1}{2}$ identical copies and each denominator correspondingly grows by a factor of $j + \frac{1}{2}$. When $j$ is an integer, the numerator splits into $j$ copies with $m > 0$ and one with $m = 0$. Recalling the factor of $1 + \delta_{m,0}$ in (3.8), the latter comes with a denominator of $2j + 1$ while all the rest come with $j + \frac{1}{2}$. We can therefore be confident that the combinatorial factors proposed below for genuine S-folds will be correct. Starting with $k = 4$, it is necessary for $j$ to be an integer. The $m = 0$ term yields $\frac{1}{2j+1}$ once again but the remaining terms are half as many in number since $m$ is required to be even.

$$
\left\langle a^{(0)} \gamma^{(1)2} \right\rangle_{a,n,\ell} = \sum_{j=0}^{\lfloor \frac{n}{2} \rfloor} \frac{2 \lfloor \frac{j}{2} \rfloor + 1}{2j + 1} \frac{\left\langle a^{(0)} \gamma^{(1)} \right\rangle_{a,n,\ell}^2}{\left\langle a^{(0)} \right\rangle_{a,n-2j,\ell}} \qquad (k = 4). \tag{4.13}
$$

Finishing with $k = 3$, half-integer values of $j$ require us to count multiples of $\frac{3}{2}$. Integer values of $j$ require us to treat $m = 0$ separately and then count multiples of 3. Putting the pieces together,

$$\left\langle a^{(0)}\gamma^{(1)2}\right\rangle_{a,n,\ell} = \left[\sum_{j=0}^{\left\lfloor\frac{n}{2}\right\rfloor}\frac{2\left\lfloor\frac{j}{3}\right\rfloor+1}{2j+1} + \sum_{j=\frac{1}{2}}^{\left\lfloor\frac{n+1}{2}\right\rfloor-\frac{1}{2}}\frac{2\left\lfloor\frac{2j}{3}\right\rfloor}{2j+1}\right]\frac{\left\langle a^{(0)}\gamma^{(1)}\right\rangle_{a,n,\ell}^2}{\left\langle a^{(0)}\right\rangle_{a,n-2j,\ell}} \qquad (k=3). \qquad (4.14)$$

One point worth making is that the information we extracted from the mixing problem was far from a full solution to it. If one wanted to know the true anomalous dimensions for instance, the matrix $M$ would have to be diagonalized. For $\mathcal{N} = 4$ SYM, this was undertaken in [64, 65] with remarkable results. The eigenvalues turned out to always be rational functions and therefore suggestive of a hidden symmetry. This was found to be a mysterious 10d conformal symmetry in [86]. As shown in [24], there is also an 8d hidden conformal symmetry applicable to the $\mathcal{N} = 2$ theories here which would be interesting to study further.[24]

## 4.2 Resumming the double discontinuity

The coefficients of conformal blocks with $\log^2 U$ in the $s$-channel are now known. Switching to the $t$-channel and using the dimension shift in table 4, the double discontinuity at one loop is given by the following sum.

$$G_a^J(z,\bar{z}) = \sum_{n=0}^{\infty}\sum_{\ell}\frac{1}{8}\left\langle a^{(0)}\gamma^{(1)2}\right\rangle_{a,n,\ell}\frac{(z-\bar{z})z^2\bar{z}^2}{(1-z)^3(1-\bar{z})^3}g_{6+2n+\ell,\ell}(1-z,1-\bar{z})\log(1-\bar{z})^2$$

$$\equiv \frac{[6G_F^{\vee}(F_t)^1_a]^2}{|G_F|(F_t)^0_a}\sum_{n=0}^{\infty}(n+1)(n+2)H_n^J(z,\bar{z})\log(1-\bar{z})^2. \qquad (4.15)$$

To iteratively compute OPE data with (2.31), one needs to be able to expand in the variables (3.1). At this point, works such as [65–67] have suggested using transcendentality methods to narrow down the special functions appearing in $G_a^J(z,\bar{z})$ until a suitable ansatz can be made. This might well be possible for all four values of $k$ but we will present a more concrete algorithm based on a key observation. *Each $H_n^J(z,\bar{z})$ is proportional to $y^n$ and therefore its contribution compared to $H_{n-1}^J(z,\bar{z})$ is suppressed by inverse powers of the spin.* In the most optimistic scenario, the powers of $y$ which can appear in a given double-twist trajectory will truncate entirely leading to a maximal value of $n$ that needs to be considered in (4.15). As such, the most important infinite sum to analyze is the one over spins. This can be understood in great generality by making use of [17].

Using the variables $x$ and $y$ (where we expand in $x$ first), the expression with explicit conformal blocks is

$$H_n^J(x,y) = \frac{x^2}{y^2}k_{n+2}\left(\frac{1}{x+1}\right)\sum_{\ell}B(n,\ell)(-1)^{n+\ell+1}k_{n+\ell+3}(-y)$$

$$- \frac{x^2}{y^2}(-1)^n k_{n+2}(-y)\sum_{\ell}B(n,\ell)k_{n+\ell+3}\left(\frac{1}{x+1}\right), \qquad (4.16)$$

where we have used a Pfaff transformation with respect to $y$. If we use a standard expansion

---

[24]The other backgrounds known to exhibit this symmetry are $AdS_3 \times S^3$ and $AdS_2 \times S^2$ [94–97]. In the former case, an analysis of unmixing was recently performed in [98].

on the easy part of (4.16), the term with $\log x$ collapses to a Legendre polynomial.[25]

$$k_{n+2}\left(\frac{1}{x+1}\right) = \frac{\Gamma(2n+4)}{\Gamma(n+2)^2}\sum_{l=0}^{\infty}\frac{(n+2)_l^2}{l!^2}\frac{x^l}{(x+1)^{l+n+2}}\left[2H_l - 2H_{n+1} - \log\left(\frac{x}{x+1}\right)\right]$$

$$= -\frac{\Gamma(2n+4)}{\Gamma(n+2)^2}P_{n+1}(2x+1)\log x + \dots. \tag{4.17}$$

It is now time to deal with the hard part of (4.16) where the sum over $\ell$ does not commute with the expansion used above. Progress will be possible because of another essential fact. *The coefficients $B(n,\ell)$ have poles at integer $\bar{h}$ and can be written as rational functions of the Casimir eigenvalue $\bar{h}(\bar{h}-1)$.* Recalling that $\bar{h} = n + \ell + 3$, this can be demonstrated with the following manipulation.

$$B(n,\ell) = \sum_j b(n,j)\frac{\bar{h}(\bar{h}-1)}{\bar{h}(\bar{h}-1)-(n+1)(n+2)}R_{2j}(\bar{h})$$

$$= \sum_j b(n,j)\bar{h}(\bar{h}-1)\prod_{l=2j+1}^{n}[\bar{h}(\bar{h}-1)-l(l+1)]R_{n+1}(\bar{h}). \tag{4.18}$$

Since (4.18) always appears beside a block, every factor of $\bar{h}(\bar{h}-1)-l(l+1)$ can be traded for $\mathcal{D}-l(l+1)$ in terms of the crossed channel Casimir

$$\mathcal{D} = (1-z)^2 z\frac{\partial^2}{\partial z^2} + (1-z)^2\frac{\partial}{\partial z} = x(x+1)\frac{\partial^2}{\partial x^2} + (2x+1)\frac{\partial}{\partial x}. \tag{4.19}$$

Up to the action of differential operators, the hard sum can therefore be done with the remarkable identities of [17]. In our notation they read

$$\sum_{\ell=0}^{\infty}\frac{R_\eta(h_0+\ell)}{\Gamma(-\eta)^2}k_{h_0+\ell}\left(\frac{1}{x+1}\right) = x^\eta + \sum_{m=0}^{\infty}\frac{\partial}{\partial m}\left[\mathcal{A}_{\eta,-m-1}^+(h_0)x^m\right],$$

$$\sum_{\ell=0}^{\infty}(-1)^\ell\frac{R_\eta(h_0+\ell)}{\Gamma(-\eta)^2}k_{h_0+\ell}\left(\frac{1}{x+1}\right) = \sum_{m=0}^{\infty}\frac{\partial}{\partial m}\left[\mathcal{A}_{\eta,-m-1}^-(h_0)x^m\right], \tag{4.20}$$

where one of the coefficients is

$$\mathcal{A}_{\eta,\zeta}^+(h_0) = -\frac{(\eta+h_0)(\zeta+h_0)}{\eta+\zeta+1}\frac{R_\eta(h_0)R_\zeta(h_0)}{\Gamma(-\eta)^2\Gamma(-\zeta)^2 r_{h_0}^2}. \tag{4.21}$$

The other one can be evaluated as a finite sum when one of its parameters is a negative integer.

$$\mathcal{A}_{\eta,-1}^-(h_0) = -(\eta+h_0)\frac{R_\eta(h_0)}{\Gamma(-\eta)^2 r_{h_0}}, \tag{4.22}$$

$$\mathcal{A}_{\eta,-m-1}^-(h_0) = \frac{\Gamma(m-\eta)^2}{\Gamma(-\eta)^2\Gamma(m+1)^2}\sum_{l=0}^{m}\frac{(\eta+1)_l(-m)_l}{(\eta-m+1)_l}\frac{(-1)^l}{l!}\mathcal{A}_{\eta-m+l,-1}^-(h_0).$$

Letting $h_0 = n+3$, there are now two limits to take associated with the sum and difference

---

[25]That the discontinuity of a single block has such a simple form has been used to great effect in [99].

Table 5: S-fold values of the coefficients $b(n,j)/R_{-2j-2}(n+2)$ that enter in the double discontinuity. They follow from (4.11), (4.12), (4.13) and (4.14) after stripping off the factors which this subsection has absorbed into the sum over spins. Note that $0 \le 2j \le n$.

|  | $j \in \mathbb{Z}$ | $j \in \mathbb{Z} + \frac{1}{2}$ |
|---|---|---|
| $k = 1$ | $(2j+1)^2$ | $(2j+1)^2$ |
| $k = 2$ | $(2j+1)^2$ | $0$ |
| $k = 3$ | $\left(2\lfloor \frac{j}{3} \rfloor + 1\right)(2j+1)$ | $2\lfloor \frac{2j}{3} \rfloor (2j+1)$ |
| $k = 4$ | $\left(2\lfloor \frac{j}{2} \rfloor + 1\right)(2j+1)$ | $0$ |

of (4.20). For the flavour representations that select even spins,

$$
\sum_{\ell^+} R_{n+1}(h_0 + \ell) k_{h_0+\ell}\left(\frac{1}{x+1}\right) = \lim_{\eta \to n+1} \Gamma(-\eta)^2 \left[\frac{1}{2}x^\eta + \sum_{m=0}^{\infty} \frac{\partial}{\partial m} \frac{\mathcal{A}^+_{\eta,-m-1}(h_0) + \mathcal{A}^-_{\eta,-m-1}(h_0)}{2\,x^{-m}}\right]
$$
$$
= \frac{x^{n+1}}{4(n+1)!^2}\left[\log^2 x + \frac{2(n+2)(H_{2n+4} - 2H_{n+1}) - 1}{n+2}\log x\right]
$$
$$
- \frac{n+2}{2(2n+3)!}\sum_{m=0}^{n} \frac{(n+3)_m(-n-1)_m}{n+1-m}\frac{(-x)^m}{m!^2}\log x
$$
$$
- \frac{1}{2}\sum_{m=0}^{\infty}\sum_{l=0}^{m} \frac{(n+2)_l(n+2-m)_l}{\Gamma(2n+4-m+l)}\frac{x^m}{l!m!}\log x + \dots, \tag{4.23}
$$

up to terms with neither $\log x$ nor $\log^2 x$. Repeating the above for odd spins, the only difference is that this sum on the last line of (4.23) changes sign. All of the ingredients are now in place for writing the final sum over spins. Dropping non-logarithmic terms again,

$$
H^J_n(x,y) = \frac{x^2}{y^2}\frac{(2n+3)!}{(n+1)!^2}P_{n+1}(2x+1)\log x \sum_{\ell^{\pm}} B(n,\ell)(-1)^{n+\ell}k_{n+\ell+3}(-y) \tag{4.24}
$$
$$
- \frac{x^2}{y^2}(-1)^n k_{n+2}(-y)\hat{B}(n,x)\left[\frac{x^{n+1}\log x(\log x + 2H_{2n+3} - 4H_{n+1})}{4(n+1)!^2} - \frac{1}{2}\sum_{m=0}^{\infty}\frac{x^m \log x}{m!}\right.
$$
$$
\left.\left(\frac{(n+2)_{m+1}(-n-1)_m}{(-1)^m m!(2n+3)!}\frac{1-\delta_{m,n+1}}{n+1-m} \pm \sum_{l=0}^{m}\frac{(n+2)_l(n+2-m)_l}{l!\Gamma(2n+4-m+l)}\right)\right] + \dots,
$$

where $B(n,\ell)$ was given in (4.18) and

$$
\hat{B}(n,x) = \sum_j b(n,j)\mathcal{D}\prod_{l=2j+1}^{n}[\mathcal{D} - l(l+1)]. \tag{4.25}
$$

The coefficients $b(n,j)$ themselves were found with the AdS unitarity method and are summarized in table 5. It will be interesting to look for further refinements to the raw formula (4.24). Some which are already apparent have been postponed to Appendix C.

## 4.3 Main results

As shown above, the resummation based on [17] is useful because it leads to an expression of the form

$$
\frac{\text{dDisc}\left[G^J_a(x,y)\right]}{4\pi^2} = \frac{[6G^\vee_F(F_t)^1_a]^2}{|G_F|(F_t)^0_a}\left[p(x,y)x^2\log^2 x + q(x,y)x^2\log x + \dots\right]. \tag{4.26}
$$

Table 6: Expressions for $S_a^{(2,2)}(n,\bar{h})/\frac{[24 G_F^\vee (F_t)_a^1]^2}{2|G_F|(F_t)_a^0}$ which is the coefficient of the triple pole in (2.31). The prefactors are $(n+1)^2(n+2)^2 r_{n+2}$.

| $k$ | $n$ | Normalized $S_a^{(2,2)}(n,\bar{h})$ |
|---|---|---|
| 1 | 0 | $2[R_0(\bar{h}) + 2R_1(\bar{h})]$ |
|   | 1 | $6[R_0(\bar{h}) + 22R_1(\bar{h}) + 120R_2(\bar{h})]$ |
|   | 2 | $\frac{36}{5}[R_0(\bar{h}) + 52R_1(\bar{h}) + 1140R_2(\bar{h}) + 10080R_3(\bar{h})]$ |
|   | 3 | $\frac{40}{7}[R_0(\bar{h}) + 92R_1(\bar{h}) + 4068R_2(\bar{h}) + 102816R_3(\bar{h}) + 1209600R_4(\bar{h})]$ |
| 2 | 0 | $2[R_0(\bar{h}) + 2R_1(\bar{h})]$ |
|   | 1 | $6[R_0(\bar{h}) + 6R_1(\bar{h}) + 24R_2(\bar{h})]$ |
|   | 2 | $\frac{36}{5}[R_0(\bar{h}) + 12R_1(\bar{h}) + 660R_2(\bar{h}) + 7200R_3(\bar{h})]$ |
|   | 3 | $\frac{40}{7}[R_0(\bar{h}) + 20R_1(\bar{h}) + 2628R_2(\bar{h}) + 50400R_3(\bar{h}) + 403200R_4(\bar{h})]$ |
| 3 | 0 | $2[R_0(\bar{h}) + 2R_1(\bar{h})]$ |
|   | 1 | $6[R_0(\bar{h}) + 6R_1(\bar{h}) + 24R_2(\bar{h})]$ |
|   | 2 | $\frac{36}{5}[R_0(\bar{h}) + 12R_1(\bar{h}) + 300R_2(\bar{h}) + 2880R_3(\bar{h})]$ |
|   | 3 | $\frac{40}{7}[R_0(\bar{h}) + 20R_1(\bar{h}) + 1116R_2(\bar{h}) + 36288R_3(\bar{h}) + 483840R_4(\bar{h})]$ |
| 4 | 0 | $2[R_0(\bar{h}) + 2R_1(\bar{h})]$ |
|   | 1 | $6[R_0(\bar{h}) + 6R_1(\bar{h}) + 24R_2(\bar{h})]$ |
|   | 2 | $\frac{36}{5}[R_0(\bar{h}) + 12R_1(\bar{h}) + 300R_2(\bar{h}) + 2880R_3(\bar{h})]$ |
|   | 3 | $\frac{40}{7}[R_0(\bar{h}) + 20R_1(\bar{h}) + 1116R_2(\bar{h}) + 20160R_3(\bar{h}) + 161280R_4(\bar{h})]$ |

Due to the structure of (4.24), the polynomials $p(x,y)$ will be the easiest ones to evaluate. Labelling them by values of $k$, we have found

$$
\begin{aligned}
p_1(x,y) = &-\frac{1}{2} - y - x(1 + 32y + 45y^2) - \frac{9}{2}x^2 y(10 + 99y + 112y^2) \\
&- 24x^3 y^2(21 + 128y + 125y^2) + O(x^4), \\
p_2(x,y) = &-\frac{1}{2} - y - x(1 + 8y + 9y^2) - \frac{9}{2}x^2 y(2 + 63y + 80y^2) \\
&- 40x^3 y^2(9 + 32y + 25y^2) + O(x^4), \\
p_3(x,y) = &-\frac{1}{2} - y - x(1 + 8y + 9y^2) - \frac{9}{2}x^2 y(2 + 27y + 32y^2) \\
&- 48x^3 y^2(3 + 24y + 25y^2) + O(x^4), \\
p_4(x,y) = &-\frac{1}{2} - y - x(1 + 8y + 9y^2) - \frac{9}{2}x^2 y(2 + 27y + 32y^2) \\
&- 16x^3 y^2(9 + 32y + 25y^2) + O(x^4).
\end{aligned}
\tag{4.27}
$$

As expected, the polynomial in $y$ beside a given power of $x$ is always finite. Further reassurance can be gained by solving for $S_a^{(2,2)}(n,\bar{h})$ and checking that it agrees with the known value of $\langle a^{(0)} \gamma^{(1)2} \rangle_{a,n,\ell}$ in accordance with (2.34). The results of this calculation are shown in table 6.

### 4.3.1 Anomalous dimension averages

Continuing to use (4.24) as a master formula inside (4.15), it is possible to solve for $q(x,y)$ as well. Together with $p(x,y)$, these polynomials permit the extraction of $S_a^{(2,1)}(n,\bar{h})$ numerators from the spectral density which contain new information. Before tabulating them, it is important to note that they will involve a discrete choice between two expressions depending on the

parity of the representation $\mathbf{R}_a$. This can be seen in the result for $\mathcal{S}_{G,1}^{(N)}$ from (4.12) which is

$$q_1(x,y) = \frac{1}{6}(-11 + 26y + 72y^2) + \frac{8}{3}x(-1 - 8y + 72y^2 + 108y^3) \tag{4.28}$$
$$+ \frac{3}{2}x^2 y(-22 + 153y + 1528y^2 + 1600y^3) + \frac{8}{3}x^3 y^2(9 + 1312y + 5375y^2 + 4500y^3)$$
$$\pm \frac{1}{6}\Big[ 5 + 34y + 36y^2 + 2x(11 + 196y + 585y^2 + 432y^3)$$
$$+ 9x^2(1 + 88y + 657y^2 + 1344y^3 + 800y^4)$$
$$- 4x^3(1 - 64y - 2079y^2 - 10560y^3 - 17400y^4 - 9000y^5)\Big] + O(x^4).$$

When $\mathbf{R}_a$ is in the symmetric (anti-symmetric) product of two adjoints, Bose symmetry requires the exchanged spin to be even (odd) which nicely comes out of the Lorentzian inversion formula. This in turn forces us to take the upper (lower) sign of (4.28) which reflects the spins that were present in the sum defining the double discontinuity.[26] After repeating the steps above for higher $\mathcal{S}_{G,1}^{(N)}$ and $\mathcal{T}_{G,1}^{(N)}$ theories, a new complication (which is familiar with maximal supersymmetry) rears its head. The analogues of (4.28) are

$$q_2(x,y) = \frac{1}{6}(-11 \pm 5) + \frac{y}{15}(-79 \pm 19) + \frac{3y^2}{35}(122 \pm 81) - \frac{8y^3}{315}(365 \pm 304) + \frac{40y^4}{693}(268 \pm 227)$$
$$+ O(y^5) + x\Big[ \frac{1}{3}(-8 \pm 11) + \frac{4y}{15}(-101 \pm 71) + \frac{3y^2}{35}(3018 \pm 1735) - \frac{16y^3}{315}(329 \pm 1123)$$
$$+ \frac{40y^4}{693}(5873 \pm 4621) + O(y^5)\Big] + x^2\Big[ \pm\frac{3}{2} + \frac{3y}{5}(-41 \pm 66) + \frac{27y^2}{70}(1245 \pm 1709)$$
$$+ \frac{4y^3}{35}(-2587 \pm 3790) + \frac{40y^4}{77}(6202 \pm 4061) + O(y^5)\Big] + x^3\Big[ \pm\frac{2}{3}(16y - 1) \tag{4.29}$$
$$+ \frac{6y^2}{35}(1276 \pm 5129) + \frac{64y^3}{315}(-4727 \pm 12242) + \frac{40y^4}{693}(265085 \pm 165103) + O(y^5)\Big] + O(x^4)$$

from (4.11),

$$q_3(x,y) = \frac{1}{6}(-11 \pm 5) + \frac{y}{15}(-79 \pm 19) + \frac{24y^2}{35}(4 \pm 3) + \frac{4y^3}{35}(20 \pm 11) - \frac{65y^4}{308}(19 \pm 14)$$
$$+ O(y^5) + x\Big[ \frac{1}{3}(-8 \pm 11) + \frac{4y}{15}(-101 \pm 71) + \frac{3y^2}{35}(678 \pm 673) + \frac{16y^3}{35}(349 \pm 185)$$
$$- \frac{5y^4}{154}(3199 \pm 2334) + O(y^5)\Big] + x^2\Big[ \pm\frac{3}{2} + \frac{3y}{5}(-41 \pm 66) + \frac{27y^2}{70}(183 \pm 755)$$
$$+ \frac{144y^3}{35}(319 \pm 212) - \frac{15y^4}{154}(-2344 \pm 287) + O(y^5)\Big] + x^3\Big[ \pm\frac{2}{3}(16y - 1) \tag{4.30}$$
$$+ \frac{6y^2}{7}(8 \pm 475) + \frac{32y^3}{21}(1429 \pm 1806) + \frac{10y^4}{231}(61232 \pm 67875) + O(y^5)\Big] + O(x^4)$$

---

[26]At the previous orders in $1/c_J$, all data depended on $(-1)^\ell$ only by virtue of whether it was zero or not. In this sense, the non-trivial effect of Bose symmetry seen here is analogous to the refined view of S-fold theory space that a one-loop calculation affords.

from (4.14) and

$$
\begin{aligned}
q_4(x,y) = {} & \frac{1}{6}(-11 \pm 5) + \frac{y}{15}(-79 \pm 19) + \frac{24y^2}{35}(4 \pm 3) - \frac{4y^3}{315}(220 \pm 191) + \frac{5y^4}{693}(953 \pm 796) \\
& + O(y^5) + x\left[\frac{1}{3}(-8 \pm 11) + \frac{4y}{15}(-101 \pm 71) + \frac{3y^2}{35}(678 \pm 673) - \frac{16y^3}{315}(209 \pm 415) \right. \\
& \left. + \frac{10y^4}{693}(13811 \pm 10411) + O(y^5)\right] + x^2\left[\pm\frac{3}{2} + \frac{3y}{5}(-41 \pm 66) + \frac{27y^2}{70}(183 \pm 755) \right. \\
& \left. + \frac{32y^3}{35}(-227 \pm 194) + \frac{10y^4}{77}(15368 \pm 9811) + O(y^5)\right] + x^3\left[\pm\frac{2}{3}(16y - 1)\right. \quad (4.31) \\
& \left. + \frac{6y^2}{7}(8 \pm 475) + \frac{64y^3}{63}(-592 \pm 1051) + \frac{40y^4}{693}(165083 \pm 94495) + O(y^5)\right] + O(x^4)
\end{aligned}
$$

from (4.13). Evidently, the spin dependence for a given twist no longer truncates which means the basic term-by-term integration advocated in Appendix A will only produce an asymptotic series valid for large spin. Iteratively finding terms up to $O(y^4)$ as in (4.29), (4.30) and (4.31) leads to the results in table 7. At the end of this subsection, we will also give results which account for all powers of $y$ and therefore go beyond asymptotics. Currently, it is only known how to do this when the value of the spin is fixed.

### 4.3.2 Non-degenerate anomalous dimensions

The entries of table 7 describe families of many dilation eigenstates which become degenerate as $c_J \to \infty$. This is of course the situation already seen at tree-level which required the infinite collection of $\langle \mathcal{O}_0 \mathcal{O}_0 \mathcal{O}_{j,m} \mathcal{O}_{j,-m}\rangle$ data summarized in (4.3). The single loop-level correlator studied here is similarly not enough to isolate a generic anomalous dimension but $[\mathcal{O}_0\mathcal{O}_0]_{0,\ell}$ is special because it is too light to mix with anything else. Gathering the $O(1)$, $O(c_J^{-1})$ and $O(c_J^{-2})$ results specific to this operator, (2.34) makes it straightforward to compute

$$
\begin{aligned}
\gamma_{a,0,\ell} = {} & \frac{G_F^\vee(F_t)^1_a}{c_J|G_F|(F_t)^0_a}\frac{24}{(\ell+1)(\ell+4)} + \left(\frac{G_F^\vee(F_t)^1_a}{c_J|G_F|(F_t)^0_a}\right)^2 \frac{96}{\ell(\ell+1)^2(\ell+4)^2(\ell+5)} \qquad (4.32) \\
& \left[\frac{4\ell^4 + 34\ell^3 + 47\ell^2 - 115\ell - 96}{(\ell+1)(\ell+4)} - \frac{5\ell^2 + 25\ell + 24}{2(-1)^\ell}\right] + O(c_T^{-1}) \qquad (k=1)
\end{aligned}
$$

for $\mathcal{S}_{G,1}^{(N)}$. As discussed above (3.1), observables of this type generically have poles which indicate a breakdown of analyticity in spin. Going back to the contour deformation in [18], sufficiently large spins were required for the arcs to drop out of the Lorentzian inversion formula. The precise condition visible in (4.32) is $\ell \geq 1$. From the perspective of the bulk, this represents a freedom to shift the amplitude by a function which satisfies three conditions.

1. Crossing symmetry

2. No double discontinuity

3. A degree compatible with gluon scattering in flat space

To enumerate such functions, it is convenient to use Mellin space and take the flat space limit of (3.24). This shows that the ambiguity is simply a constant or a Mack polynomial of spin zero.[27] As further confirmation, the Mellin amplitude found in [44] is a formal expression which

---

[27]See [72, 75] for an explanation of how such terms may be fixed with localization.

Table 7: Expressions for $S_a^{(2,1)}(n,\bar{h})/\frac{[24G_F^\vee(F_t)^1{}_a]^2}{2|G_F|(F_t)^0{}_a}$ which is the coefficient of the double pole in (2.31). For $k=1$ they have been found exactly while for $k>1$ we give the first five terms in the asymptotic series which is valid for large $\bar{h}(\bar{h}-1)$. This time, the appropriate formula depends on whether it is being used to compute an even-spin or odd-spin trajectory.

| $k$ | $\ell$ | $n$ | Normalized $S_a^{(2,1)}(n,\bar{h})$ |
|---|---|---|---|
| 1 | $+$ | 0 | $R_0(\bar{h})-10R_1(\bar{h})-72R_2(\bar{h})$ |
| | | 1 | $\frac{1}{2}[R_0(\bar{h})-106R_1(\bar{h})-3240R_2(\bar{h})-31104R_3(\bar{h})]$ |
| | | 2 | $-\frac{3}{25}[7R_0(\bar{h})+1149R_1(\bar{h})+58710R_2(\bar{h})+1486800R_3(\bar{h})+17280000R_4(\bar{h})]$ |
| 1 | $-$ | 0 | $\frac{4}{3}[2R_0(\bar{h})+R_1(\bar{h})-18R_2(\bar{h})]$ |
| | | 1 | $\frac{1}{2}[19R_0(\bar{h})+178R_1(\bar{h})-24R_2(\bar{h})-10368R_3(\bar{h})]$ |
| | | 2 | $\frac{3}{25}[118R_0(\bar{h})+2741R_1(\bar{h})+26730R_2(\bar{h})-147600R_3(\bar{h})-5760000R_4(\bar{h})]$ |
| 2 | $+$ | 0 | $\frac{1}{35}[35R_0(\bar{h})+140R_1(\bar{h})-2436R_2(\bar{h})-21408R_3(\bar{h})-576000R_4(\bar{h})]+\dots$ |
| | | 1 | $\frac{1}{70}[35R_0(\bar{h})+910R_1(\bar{h})-118944R_2(\bar{h})+143040R_3(\bar{h})-25574400R_4(\bar{h})]+\dots$ |
| | | 2 | $-\frac{3}{175}[49R_0(\bar{h})-497R_1(\bar{h})+408786R_2(\bar{h})+77808R_3(\bar{h})+210240000R_4(\bar{h})]+\dots$ |
| 2 | $-$ | 0 | $\frac{2}{1155}[1540R_0(\bar{h})+3773R_1(\bar{h})-8118R_2(\bar{h})-32208R_3(\bar{h})-787200R_4(\bar{h})]+\dots$ |
| | | 1 | $\frac{1}{770}[7315R_0(\bar{h})+41118R_1(\bar{h})-349536R_2(\bar{h})-1160896R_3(\bar{h})-33100800R_4(\bar{h})]+\dots$ |
| | | 2 | $\frac{3}{1925}[9086R_0(\bar{h})+93709R_1(\bar{h})+53658R_2(\bar{h})+15416368R_3(\bar{h})-451660800R_4(\bar{h})]+\dots$ |
| 3 | $+$ | 0 | $\frac{1}{35}[35R_0(\bar{h})+140R_1(\bar{h})-672R_2(\bar{h})-4464R_3(\bar{h})+140400R_4(\bar{h})]+\dots$ |
| | | 1 | $\frac{1}{70}[35R_0(\bar{h})+910R_1(\bar{h})-33768R_2(\bar{h})+606240R_3(\bar{h})+7524000R_4(\bar{h})]+\dots$ |
| | | 2 | $-\frac{3}{175}[49R_0(\bar{h})-497R_1(\bar{h})+123522R_2(\bar{h})+3823344R_3(\bar{h})-2462400R_4(\bar{h})]+\dots$ |
| 3 | $-$ | 0 | $\frac{2}{1155}[1540R_0(\bar{h})+3773R_1(\bar{h})-1584R_2(\bar{h})-21384R_3(\bar{h})+351000R_4(\bar{h})]+\dots$ |
| | | 1 | $\frac{1}{770}[7315R_0(\bar{h})+41118R_1(\bar{h})-3432R_2(\bar{h})+2049696R_3(\bar{h})+12924000R_4(\bar{h})]+\dots$ |
| | | 2 | $\frac{3}{1925}[9086R_0(\bar{h})+93709R_1(\bar{h})+586674R_2(\bar{h})-7585776R_3(\bar{h})-78912000R_4(\bar{h})]+\dots$ |
| 4 | $+$ | 0 | $\frac{1}{35}[35R_0(\bar{h})+140R_1(\bar{h})-672R_2(\bar{h})+6576R_3(\bar{h})-254400R_4(\bar{h})]+\dots$ |
| | | 1 | $\frac{1}{70}[35R_0(\bar{h})+910R_1(\bar{h})-33768R_2(\bar{h})+93024R_3(\bar{h})-14601600R_4(\bar{h})]+\dots$ |
| | | 2 | $-\frac{3}{175}[49R_0(\bar{h})-497R_1(\bar{h})+123522R_2(\bar{h})-169776R_3(\bar{h})+127742400R_4(\bar{h})]+\dots$ |
| 4 | $-$ | 0 | $\frac{2}{1155}[1540R_0(\bar{h})+3773R_1(\bar{h})-1584R_2(\bar{h})+7656R_3(\bar{h})-376800R_4(\bar{h})]+\dots$ |
| | | 1 | $\frac{1}{770}[7315R_0(\bar{h})+41118R_1(\bar{h})-3432R_2(\bar{h})-279840R_3(\bar{h})-22262400R_4(\bar{h})]+\dots$ |
| | | 2 | $\frac{3}{1925}[9086R_0(\bar{h})+93709R_1(\bar{h})+586674R_2(\bar{h})+8534064R_3(\bar{h})-294187200R_4(\bar{h})]+\dots$ |

does not converge until one chooses a regularization scheme which leaves the same contact term unfixed. Finally, the double discontinuity in (2.31) also appears in a newer dispersion relation [100] which directly reconstructs a four-point function (either in position or Mellin space) instead of the spectral density. In this language, the ambiguities at low spin manifest themselves as subtractions.

Proceeding to higher values of $k$ reveals a piece of good fortune. Since the double-trace $\left[\mathcal{O}_{\frac{1}{2}}\mathcal{O}_{\frac{1}{2}}\right]_{0,\ell}$ does not exist in any of the genuine S-folds, we can actually go up to $n=1$. Defining $\tilde{c}_{J,a}=c_J/\frac{G_F^\vee(F_t)^1{}_a}{|G_F|(F_t)^0{}_a}$ for convenience, the same analysis leads to

$$\gamma_{a,0,\ell}=\frac{24}{\tilde{c}_{J,a}(\ell+1)(\ell+4)}+\frac{96}{35\tilde{c}_{J,a}^2(\ell)_6(\ell+1)(\ell+4)}\tag{4.33}$$

$$\left[\frac{140\ell^6+1890\ell^5+9443\ell^4+21420\ell^3+16265\ell^2-17550\ell-17568}{(\ell+1)(\ell+4)}\right.$$

$$\left.-\frac{(-1)^\ell}{2}(175\ell^4+1750\ell^3+5341\ell^2+4830\ell+5832)-\frac{96(365+304(-1)^\ell)}{(\ell-1)(\ell+6)}+\dots\right]$$

$$\gamma_{a,1,\ell} = \frac{72}{\tilde{c}_{J,a}(\ell+1)(\ell+6)} + \frac{216}{35\tilde{c}_{J,a}^2(\ell+1)_6(\ell+1)(\ell+6)} \qquad (k=2)$$

$$\left[ \frac{945\ell^6 + 19005\ell^5 + 150493\ell^4 + 591227\ell^3 + 1015438\ell^2 - 92148\ell - 699192}{(\ell+1)(\ell+6)} \right.$$

$$\left. -(-1)^\ell(525\ell^4 + 7350\ell^3 + 36953\ell^2 + 78596\ell + 135636) + \frac{128(293 + 1942(-1)^\ell)}{\ell(\ell+7)} + \cdots \right]$$

for $\mathcal{S}_{G,2}^{(N)}$ or $\mathcal{T}_{G,2}^{(N)}$,

$$\gamma_{a,0,\ell} = \frac{24}{\tilde{c}_{J,a}(\ell+1)(\ell+4)} + \frac{96}{35\tilde{c}_{J,a}^2(\ell)_6(\ell+1)(\ell+4)} \qquad (4.34)$$

$$\left[ \frac{140\ell^6 + 1890\ell^5 + 9443\ell^4 + 21420\ell^3 + 19505\ell^2 - 1350\ell - 4608}{(\ell+1)(\ell+4)} \right.$$

$$\left. -\frac{(-1)^\ell}{2}(175\ell^4 + 1750\ell^3 + 5341\ell^2 + 4830\ell + 1728) - \frac{432(20 + 11(-1)^\ell)}{(\ell-1)(\ell+6)} + \cdots \right]$$

$$\gamma_{a,1,\ell} = \frac{72}{\tilde{c}_{J,a}(\ell+1)(\ell+6)} + \frac{864}{35\tilde{c}_{J,a}^2(\ell+1)_6(\ell+1)(\ell+6)} \qquad (k=3)$$

$$\left[ \frac{210\ell^6 + 4200\ell^5 + 33187\ell^4 + 130718\ell^3 + 250627\ell^2 + 152418\ell - 9288}{(\ell+1)(\ell+6)} \right.$$

$$\left. -\frac{(-1)^\ell}{2}(315\ell^4 + 4410\ell^3 + 21889\ell^2 + 45178\ell + 44112) + \frac{144(1376 + 729(-1)^\ell)}{\ell(\ell+7)} + \cdots \right]$$

for $\mathcal{S}_{G,3}^{(N)}$ or $\mathcal{T}_{G,3}^{(N)}$ and

$$\gamma_{a,0,\ell} = \frac{24}{\tilde{c}_{J,a}(\ell+1)(\ell+4)} + \frac{96}{35\tilde{c}_{J,a}^2(\ell)_6(\ell+1)(\ell+4)} \qquad (4.35)$$

$$\left[ \frac{140\ell^6 + 1890\ell^5 + 9443\ell^4 + 21420\ell^3 + 19505\ell^2 - 1350\ell - 4608}{(\ell+1)(\ell+4)} \right.$$

$$\left. -\frac{(-1)^\ell}{2}(175\ell^4 + 1750\ell^3 + 5341\ell^2 + 4830\ell + 1728) + \frac{48(220 + 191(-1)^\ell)}{(\ell-1)(\ell+6)} + \cdots \right]$$

$$\gamma_{a,1,\ell} = \frac{72}{\tilde{c}_{J,a}(\ell+1)(\ell+6)} + \frac{864}{35\tilde{c}_{J,a}^2(\ell+1)_6(\ell+1)(\ell+6)} \qquad (k=4)$$

$$\left[ \frac{210\ell^6 + 4200\ell^5 + 33187\ell^4 + 130718\ell^3 + 250627\ell^2 + 152418\ell - 9288}{(\ell+1)(\ell+6)} \right.$$

$$\left. -\frac{(-1)^\ell}{2}(315\ell^4 + 4410\ell^3 + 21889\ell^2 + 45178\ell + 44112) + \frac{48(352 + 617(-1)^\ell)}{\ell(\ell+7)} + \cdots \right]$$

for $\mathcal{S}_{G,4}^{(N)}$ or $\mathcal{T}_{G,4}^{(N)}$. Less fortunately, the finite spin regime is not controlled so the poles at positive values of the spin are not meaningful. Indeed, they would disappear if one re-expanded the anomalous dimensions in $1/\ell$ and regarded the suppressed terms as $O(\ell^{-12})$ corrections. Nevertheless, we will soon see that the Lorentzian inversion formula without the arcs still converges for $\ell \geq 1$. This is inevitable because the value of $k$ cannot affect the growth of the $\langle \mathcal{O}_0\mathcal{O}_0\mathcal{O}_0\mathcal{O}_0 \rangle$ correlator in the flat space limit.

### 4.3.3 Some finite spin completions

To find symbolic expressions for S-fold anomalous dimensions at finite spin, one needs the double discontinuity to be exact in $y$ at a given order in $x$. Below we present this even more

resummed version of the double discontinuity for the case of $k = 2$. This formula, which is conjectural, came from computing terms up to $O(y^{50})$ in (4.29) and then using Mathematica to discern a pattern in the coefficients.[28] For even spins, the result is

$$q_2(z,\bar{z}) = \frac{225\bar{z}^5 - 30\bar{z}^4 + 16\bar{z}^3 + 1392\bar{z}^2 - 4160\bar{z} + 2304}{256\bar{z}^2} \tag{4.36}$$
$$+ \frac{225\bar{z}^5 - 180\bar{z}^4 + 16\bar{z}^3 - 64\bar{z}^2 + 512\bar{z} - 512}{256\bar{z}} \frac{\text{atanh}(\sqrt{1-\bar{z}})}{\sqrt{1-\bar{z}}}$$
$$+ \frac{z}{1-z}\left[ \frac{11025\bar{z}^7 - 4800\bar{z}^6 - 276\bar{z}^5 + 704\bar{z}^4 - 25344\bar{z}^3 + 233984\bar{z}^2 - 435456\bar{z} + 221184}{1024\bar{z}^3} \right.$$
$$+ \frac{11025\bar{z}^7 - 12150\bar{z}^6 + 1944\bar{z}^5 + 848\bar{z}^4 - 3968\bar{z}^3 + 52480\bar{z}^2 - 105472\bar{z} + 55296}{1024\bar{z}^2}$$
$$\left. \frac{\text{atanh}(\sqrt{1-\bar{z}})}{\sqrt{1-\bar{z}}} \right] + O(z^2)$$

and for odd spins, it is

$$q_2(z,\bar{z}) = \frac{135\bar{z}^5 - 90\bar{z}^4 + 3136\bar{z}^3 - 10800\bar{z}^2 + 9920\bar{z} - 2304}{-768\bar{z}^2(1-\bar{z})} \tag{4.37}$$
$$+ \frac{45\bar{z}^5 - 60\bar{z}^4 - 48\bar{z}^3 + 576\bar{z}^2 - 1024\bar{z} + 512}{-768\bar{z}(1-\bar{z})} \frac{3\,\text{atanh}(\sqrt{1-\bar{z}})}{\sqrt{1-\bar{z}}}$$
$$+ \frac{z}{1-z}\left[ \frac{6075\bar{z}^7 - 3780\bar{z}^6 - 180\bar{z}^5 - 43456\bar{z}^4 + 410496\bar{z}^3 - 947456\bar{z}^2 + 799488\bar{z} - 221184}{-3072\bar{z}^3(1-\bar{z})} \right.$$
$$+ \frac{2025\bar{z}^7 - 2610\bar{z}^6 + 600\bar{z}^5 - 3984\bar{z}^4 + 56448\bar{z}^3 - 157952\bar{z}^2 + 160768\bar{z} - 55296}{-3072\bar{z}^2(1-\bar{z})}$$
$$\left. \frac{3\,\text{atanh}(\sqrt{1-\bar{z}})}{\sqrt{1-\bar{z}}} \right] + O(z^2).$$

Although it would be nice to treat $k = 3$ and $k = 4$ in the same way, these apparently lead to coefficient sequences which are not known to Mathematica.

We can now attempt to integrate conformal blocks against (4.36) and (4.37) for any value of the twist. It will be most interesting to do this for the two non-degenerate families thereby improving (4.33). As discussed in [75], one also needs to set the spin to a certain integer for the result to have a closed form. Picking one even example and one odd example (and recalling that $\ell = 0$ is ambiguous due to the aforementioned contact term), it follows that

$$\gamma_{a,0,1} = \frac{12}{5\tilde{c}_{J,a}} + \frac{3}{\tilde{c}_{J,a}^2} \frac{-101126 + 353280\log(2) - 127575\zeta(3)}{4000}, \tag{4.38}$$
$$\gamma_{a,0,2} = \frac{4}{3\tilde{c}_{J,a}} + \frac{1}{\tilde{c}_{J,a}^2} \frac{-9586834 + 35868672\log(2) - 12720645\zeta(3)}{14112},$$
$$\gamma_{a,1,1} = \frac{36}{7\tilde{c}_{J,a}} + \frac{3}{\tilde{c}_{J,a}^2} \frac{-56280862 + 209950720\log(2) - 74473875\zeta(3)}{137200},$$
$$\gamma_{a,1,2} = \frac{3}{\tilde{c}_{J,a}} + \frac{1}{\tilde{c}_{J,a}^2} \frac{-9784676042 + 36663459848\log(2) - 13001749425\zeta(3)}{702464}. \tag{$k = 2$}$$

The anomalous dimensions in this section, whether resummed or asymptotic, can be made fully explicit by plugging in a central charge from table 3 and a crossing matrix from Appendix B. There is also the important matter of supergravity at order $1/c_T$ to which we now turn.

---

[28]More precisely, we replaced $y$ with $\frac{1-\bar{z}}{\bar{z}}$ again and then expanded around $\bar{z} = 1$. Before doing so, it was also necessary to introduce an overall prefactor of $1/\bar{z}$. We thank Tobias Hansen for suggesting this way of doing the calculation.

## 4.4 The supergravity contribution

Going back to the data from the brane construction, $c_J^2$ and $c_T$ are both proportional to $N^2$. The double discontinuity $G^J(z, \bar{z})$ should therefore be considered alongside a similar function $G^T(z, \bar{z})$ from tree-level supergravity. The Mellin formalism is once again a convenient tool for computing it. In addition to agreeing with the position space calculation of [44], this will allow us to formalize the open problem of repeating it for heavier external operators.

### 4.4.1 Lightest external operator

The first step is to take the $\mathcal{N} = 4$ multiplets for $AdS_5 \times S^5$ and check which ones contain $\mathcal{N} = 2$ multiplets that are allowed by the $\mathcal{O}_0 \times \mathcal{O}_0$ OPE. As expected, only the lowest Kaluza-Klein mode does. Its decomposition was first found in [101] and reads

$$B\bar{B}[0;0]_2^{(0,2,0)} = B\bar{B}[0;0]_2^{(2,2,0)} \oplus A\bar{A}[0;0]_2^{(0,0,0)} \qquad (4.39)$$
$$\oplus A\bar{B}[0;0]_2^{(1,1,2)} \oplus B\bar{A}[0;0]_2^{(1,1,-2)} \oplus L\bar{B}[0;0]_2^{(0,0,4)} \oplus B\bar{L}[0;0]_2^{(0,0,-4)},$$

where we have switched to the notation of [79]. The extra Dynkin label on the right-hand side is for keeping track of $SU(2)_L$ which is the commutant of $SU(2)_R \times U(1)_R$ in $SO(6)_R$. Notice that only one of the six multiplets in (4.39) is exchanged because the others have too much charge under either $U(1)_R$ or $SU(2)_L$. Based on this, the appropriate Mellin space ansatz is

$$\mathcal{M}^{I_1 I_2 I_3 I_4}(s, t; \alpha) = \delta^{I_1 I_2} \delta^{I_3 I_4} \left[ \lambda_0^{A\bar{A}} \mathcal{S}_0^{A\bar{A}}(s, t; \alpha) + \mathcal{C}(s, t; \alpha) \right] + \dots, \qquad (4.40)$$

where only the $s$-channel has been written explicitly. Our goal is to apply the superconformal Ward identity to this piece alone and then write down the orbit of the resulting solution under crossing.[29] This means using $\{\delta^{I_1 I_2} \delta^{I_3 I_4}, \delta^{I_1 I_4} \delta^{I_2 I_3}, \delta^{I_1 I_3} \delta^{I_2 I_4}\}$ (which are linearly independent for almost all groups) as basis elements instead of projections onto irreducible representations in a fixed channel. The calculation can be setup by looking up the explicit superconformal block for the stress tensor multiplet [85] which gives

$$\mathcal{S}_0^{A\bar{A}}(s, t; \alpha) = \mathcal{Y}_0(\alpha) \mathcal{M}_{2,0}(s, t) - \mathcal{Y}_1(\alpha) \mathcal{M}_{3,1}(s, t) + \frac{1}{15} \mathcal{Y}_0(\alpha) \mathcal{M}_{4,2}(s, t) \qquad (4.41)$$
$$= \sum_{m=0}^{\infty} \frac{-3[\alpha(t-u)(t+u-10) - (u-4)(t-u+2)]}{m! \Gamma[1-m]^2 \Gamma[m+3](s-2-2m)}.$$

In the second line, we have made contact with [19, 20] by solving for the $s = 2 + 2m$ residue and setting $2m = 6 - t - u$ in the polynomial part. This shows the expected zero in the limit of maximal R-symmetry violation (MRV). Another useful check of (4.41) can be performed with the superconformal twist (which does not see contact terms). The only $s$-channel singularity which could potentially contribute to the chiral correlator for (4.40) comes from the pole at $s = 2$. The integral

$$f(z, \bar{z}; \bar{z}^{-1}) \propto \int_{-i\infty}^{o\infty} \frac{dt}{2\pi i} UV^{\frac{t}{2}-2} \left[ (t-2)^2 - 4\bar{z}^{-1}(t-3) \right] \Gamma\left( \frac{4-t}{2} \right)^2 \Gamma\left( \frac{t-2}{2} \right)^2 \qquad (4.42)$$

can then be evaluated along the lines of [90]. With $\frac{\partial f}{\partial \bar{z}} = 0$ guaranteed, there is no loss of generality in setting $\bar{z} = 1$ which localizes the integral to the neighbourhood of $t = 4$. Seeing

---

[29]Strictly speaking, this adds one assumption to our previous approach. If the single-trace spectrum were the only dynamical input, this would not say anything about the $1/c_T$ and $1/c_J^2$ contributions to a four-point function posessing superconformal symmetry individually.

that there is no pole here, we confirm the fact that (3.10) is an exact four-point function of affine currents.[30]

Returning to the evaluation of $\langle \mathcal{O}_0 \mathcal{O}_0 \mathcal{O}_0 \mathcal{O}_0 \rangle$, it is clear that the contact term $\mathcal{C}(s, t; \alpha)$ has degree 2 in $\alpha$ and degree 1 in the Mandelstam variables. Letting the coefficients of these powers be arbitrary, we can apply the superconformal Ward identity to (4.40) and arrive at

$$\mathcal{C}(s, t; \alpha) = \frac{3\lambda_0^{A\bar{A}}}{8} \left[ -8 + 2(2\alpha - 1)(t - u) + 2(2\alpha - 1)^2 (t + u - 4) \right],  \tag{4.43}$$

which is manifestly Bose symmetric. By checking how the stress tensor appears, the overall normalization is $\lambda_0^{A\bar{A}} = -\frac{80}{3c_T}$. To obtain an auxiliary Mellin amplitude by inverting (3.24), one can employ the "triangle method" of [10] which becomes trivial when there is only one R-symmetry cross ratio. As in [44],

$$\widetilde{\mathcal{M}}^{I_1 I_2 I_3 I_4}(s, t) = -\frac{320}{c_T} \left[ \frac{\delta^{I_1 I_2} \delta^{I_3 I_4}}{s - 2} + \frac{\delta^{I_1 I_4} \delta^{I_2 I_3}}{t - 2} + \frac{\delta^{I_1 I_3} \delta^{I_2 I_4}}{\tilde{u} - 2} \right],  \tag{4.44}$$

which makes it easy to extract the double discontinuity. Following (4.64),

$$\begin{aligned}
G_a^T(U, V) &= -\frac{|G_F|(F_t)^0_a}{c_T} \frac{80}{V} \int_{-i\infty}^{i\infty} \frac{ds}{2\pi i} U^{\frac{s}{2}} \Gamma\left(2 - \frac{s}{2}\right)^2 \Gamma\left(\frac{s}{2}\right)^2 \\
&= -\frac{|G_F|(F_t)^0_a}{3c_T} \frac{80}{V} {}_2F_1\left(2, 2; 4; 1 - U^{-1}\right)
\end{aligned}  \tag{4.45}$$

and the spectral density goes as $\frac{1}{2} \csc^2[\pi(3 - h)] R_{-3}(h) R_{-1}(\bar{h}) + \csc^2[\pi(2 - h)] R_{-2}(h) R_{-1}(\bar{h})$. This function serves to bring about the additional term

$$\begin{aligned}
\left\langle a^{(0)} \gamma^{(2)} \right\rangle_{a,n,\ell} &\mapsto \left\langle a^{(0)} \gamma^{(2)} \right\rangle_{a,n,\ell} \\
&+ \frac{160 |G_F|(F_t)^0_a}{c_T} \left[ \frac{1}{2} R_{-3}(n + 2) + R_{-2}(n + 2) \right] R_{-1}(n + \ell + 3),
\end{aligned}  \tag{4.46}$$

for every one-loop anomalous dimension that was computed from $G_a^J(U, V)$ in the previous subsection. As with (3.28), this correction is insensitive to the S-fold construction because it comes from tree-level dynamics.

### 4.4.2 Comments on higher weights

Encouraged by this result, a logical next step is to try fixing the $1/c_T$ part of $\left\langle \mathcal{O}_{\frac{1}{2}} \mathcal{O}_{\frac{1}{2}} \mathcal{O}_{\frac{1}{2}} \mathcal{O}_{\frac{1}{2}} \right\rangle$ consisting of dimension 3 primaries. These are allowed to couple to various components of the $\frac{1}{2}$-BPS $\mathcal{N} = 4$ multiplet of dimension 4.

$$\begin{aligned}
B\bar{B}[0; 0]_4^{(0,4,0)} = {}&B\bar{B}[0; 0]_4^{(4,4,0)} \oplus A\bar{A}[0; 0]_4^{(2,2,0)} \oplus L\bar{L}[0; 0]_4^{(0,0,0)} \\
&\oplus A\bar{B}[0; 0]_4^{(3,3,2)} \oplus B\bar{A}[0; 0]_4^{(3,3,-2)} \oplus L\bar{B}[0; 0]_4^{(0,0,8)} \oplus B\bar{L}[0; 0]_4^{(0,0,-8)} \oplus \\
&\bigoplus_{r=1}^{2} \left[ L\bar{B}[0; 0]_4^{(r,r,8-2r)} \oplus B\bar{L}[0; 0]_4^{(r,r,2r-8)} \right] \oplus \bigoplus_{r=0}^{1} \left[ L\bar{A}[0; 0]_4^{(r,r,4-2r)} \oplus A\bar{L}[0; 0]_4^{(r,r,2r-4)} \right].
\end{aligned}  \tag{4.47}$$

---

[30]This does not contradict the fact that the stress tensor multiplet contains a Schur operator. The double-trace multiplet $B\bar{B}[0; 0]_4^{(4,0)}$ also does and the two become degenerate in the chiral algebra. The cancellation between them underlies a 4d unitarity bound derived in [89].

Due to the long $\mathcal{N} = 2$ multiplet on the right-hand side, we should no longer expect the superconformal Ward identity to fix everything. This can be verified by using the ansatz

$$\mathcal{M}^{I_1 I_2 I_3 I_4}(s, t; \alpha, \beta) = \delta^{I_1 I_2} \delta^{I_3 I_4} \left[ \lambda_0^{A\bar{A}} \mathcal{S}_0^{A\bar{A}}(s, t; \alpha) + \lambda_0^{L\bar{L}} \mathcal{S}_0^{L\bar{L}}(s, t; \alpha) + \mathcal{C}_0(s, t; \alpha) \right] \quad (4.48)$$
$$+ \delta^{I_1 I_2} \delta^{I_3 I_4} \mathcal{Y}_1(\beta) \left[ \lambda_1^{B\bar{B}} \mathcal{S}_1^{B\bar{B}}(s, t; \alpha) + \lambda_1^{A\bar{A}} \mathcal{S}_1^{A\bar{A}}(s, t; \alpha) + \mathcal{C}_1(s, t; \alpha) \right]$$

and again writing down explicit blocks in the $s$-channel. The new ones are

$$\mathcal{S}_1^{A\bar{A}}(s, t; \alpha) = \mathcal{Y}_1(\alpha) \mathcal{M}_{4,0}(s, t) - \left[ \mathcal{Y}_2(\alpha) + \frac{1}{12} \mathcal{Y}_0(\alpha) \right] \mathcal{M}_{5,1}(s, t)$$
$$+ \frac{9}{140} \mathcal{Y}_1(\alpha) \mathcal{M}_{6,2}(s, t) + \frac{1}{12} \mathcal{Y}_1(\alpha) \mathcal{M}_{6,0}(s, t) - \frac{3}{560} \mathcal{Y}_0(\alpha) \mathcal{M}_{7,1}(s, t),$$
$$\mathcal{S}_0^{L\bar{L}}(s, t; \alpha) = \mathcal{Y}_0(\alpha) \mathcal{M}_{4,0}(s, t) - \mathcal{Y}_1(\alpha) \mathcal{M}_{5,1}(s, t) + \left[ \mathcal{Y}_2(\alpha) + \frac{1}{60} \mathcal{Y}_0(\alpha) \right] \mathcal{M}_{6,0}(s, t)$$
$$+ \frac{9}{140} \mathcal{Y}_0(\alpha) \mathcal{M}_{6,2}(s, t) - \frac{9}{140} \mathcal{Y}_1(\alpha) \mathcal{M}_{7,1}(s, t) + \frac{3}{700} \mathcal{Y}_0(\alpha) \mathcal{M}_{8,0}(s, t), \quad (4.49)$$

which show that the R-symmetry representation for a given Witten diagram need not be unique. These multiplets also contain twists of 6 and 8 which have vanishing residues because they overlap with double-trace values. The corresponding Witten diagrams are therefore pure contact terms which allow the contributions with lower twists to be symmetrized. This leads to the residues

$$\mathcal{S}_0^{A\bar{A}}(s, t; \alpha) = \sum_{m=0}^{\infty} \frac{-3[\alpha(t-u)(t+u-14) - (u-6)(t-u+2)]}{m! \Gamma[2-m]^2 \Gamma[m+3](s-2-2m)}, \quad (4.50)$$
$$\mathcal{S}_1^{A\bar{A}}(s, t; \alpha) = \sum_{m=0}^{\infty} \frac{-60}{4m! \Gamma[1-m]^2 \Gamma[m+5](s-4-2m)}$$
$$\left[ 3\alpha^2(t-u)(t+u-16) - \alpha(t^2+2tu-32t-5u^2+64u-120) - (u-6)(t-2u+10) \right],$$
$$\mathcal{S}_0^{L\bar{L}}(s, t; \alpha) = \sum_{m=0}^{\infty} \frac{-90}{m! \Gamma[1-m]^2 \Gamma[m+5](s-4-2m)}$$
$$\left[ \alpha^2(t+u-16)(t+u-8) - 2\alpha(tu-6t+u^2-18u+64) + (u-6)^2 \right],$$

where we have restated $\mathcal{S}_0^{A\bar{A}}$ since the external weights are now different.[31] One should similarly analyze $\mathcal{S}_1^{B\bar{B}}$ (which was called $\mathcal{S}_0$ in (3.18)) but this was already done in [24]. Using the superconformal Ward identity once again, there are indeed more free parameters than before. Nevertheless, the solution

$$\lambda_1^{B\bar{B}} = 0, \quad \lambda_0^{A\bar{A}} = 0, \quad \mathcal{C}_0(s, t; \alpha) = \frac{75\lambda_0^{L\bar{L}}}{8} [(2\alpha-1)^2 - 1] \quad (4.51)$$
$$\mathcal{C}_1(s, t; \alpha) = \frac{5\lambda_1^{A\bar{A}}}{64} \left[ u - t - (2\alpha-1)(t+u+6) + 3(2\alpha-1)^2(t-u) + 3(2\alpha-1)^3(t+u-6) \right]$$

reveals a surprising absence of the $1/c_T$ term $\lambda_0^{A\bar{A}}$. Understanding this issue will be important for improving the status of massive loop calculations with eight supercharges.

---

[31] We do not have an explanation for why the long block has a double zero instead of a single zero in the MRV limit. While this pattern continues to hold for higher weights, it appears to be unique to four dimensions.

## 4.5 Matching onto a Mellin amplitude

Although we have gone this far by only using Mellin space for tree-level calculations, there is also an algorithm [68] for building a one-loop Mellin amplitude once its double discontinuity is known. For reasons that are not yet understood, it is often the case that this amplitude can be chosen to have only simultaneous poles and no simple poles. This allows its structure to be determined solely from the simplest part of the double discontinuity which was called $p(x, y)$ in (4.26). To see how this works, let us quote

$$p_k(x, y) = \sum_{m=0}^{\infty} x^m y^{m-1} \left( \frac{m+1}{2} \right)^2$$
$$[T_k(m)(m+2)y(my + 2y + m + 1) + T_k(m-1)m(my + y + m)], \quad (4.52)$$

where

$$T_1(2j) = \frac{1}{3}(j+1)(2j+1)(4j+3), \quad (4.53)$$

$$T_2(2j) = \frac{1}{3}(\lfloor j \rfloor + 1)(2\lfloor j \rfloor + 1)(2\lfloor j \rfloor + 3),$$

$$T_3(2j) = \left( \lfloor \tfrac{j}{3} \rfloor + 1 \right) \left( 4\lfloor \tfrac{j}{3} \rfloor^2 + 6\lfloor \tfrac{j}{3} \rfloor + 1 \right) + 8\lfloor \tfrac{2j-1}{6} \rfloor \left( \lfloor \tfrac{2j-1}{6} \rfloor + 1 \right)^2$$
$$+ \left( \lfloor \tfrac{j-1}{3} \rfloor + 1 \right) \left( 2\lfloor \tfrac{j-1}{3} \rfloor + 1 \right) \left( 2\lfloor \tfrac{j-1}{3} \rfloor + 3 \right) + 2 \left( \lfloor \tfrac{2j-3}{6} \rfloor + 1 \right) \left( 4\lfloor \tfrac{2j-3}{6} \rfloor^2 + 9\lfloor \tfrac{2j-3}{6} \rfloor + 4 \right)$$
$$+ \left( \lfloor \tfrac{j-2}{3} \rfloor + 1 \right) \left( 4\lfloor \tfrac{j-2}{3} \rfloor^2 + 10\lfloor \tfrac{j-2}{3} \rfloor + 5 \right) + 2 \left( \lfloor \tfrac{2j-5}{6} \rfloor + 1 \right) \left( \lfloor \tfrac{2j-5}{6} \rfloor + 2 \right) \left( 4\lfloor \tfrac{2j-5}{6} \rfloor + 3 \right),$$

$$T_4(2j) = \frac{1}{3} \left( \lfloor \tfrac{j-1}{2} \rfloor + 1 \right) \left( 8\lfloor \tfrac{j-1}{2} \rfloor^2 + 19\lfloor \tfrac{j-1}{2} \rfloor + 9 \right) + \frac{1}{3} \left( \lfloor \tfrac{j}{2} \rfloor + 1 \right) \left( 8\lfloor \tfrac{j}{2} \rfloor^2 + 13\lfloor \tfrac{j}{2} \rfloor + 3 \right).$$

These can be readily derived from the results in Appendix C. There will be no need to perform the sum here, but for $T_k$ coefficients of the form (4.53), the result is always a rational function of $x$ and $y$. Switching back to the cross ratios $U$ and $V$, (4.52) tells us all terms in the four-point function which include a factor of $\log^2 U \log^2 V$. These naturally lead to simultaneous poles when we go to Mellin space. Specifically,

$$\widetilde{\mathcal{M}}(s, t) = \frac{1}{(s-2l)(t-2m)} \Rightarrow H(U, V) = \frac{\Gamma(l+m-1)^2 U^l V^{m-2}}{16\Gamma(l-1)^2 \Gamma(m-1)^2} \log^2 U \log^2 V + \dots, \quad (4.54)$$

where the integers $l$ and $m$ are at least 2. Taking a linear combination of such poles and making the result crossing symmetric (anti-symmetric) for even (odd) $\mathbf{R}_a$,

$$\widetilde{\mathcal{M}}_a(s, t) = \frac{[6G_F^\vee (F_t)^1_a]^2}{c_J^2 |G_F| (F_t)^0_a} \sum_{l,m=2}^{\infty} c_{lm} \left[ \frac{1}{(s-2l)(t-2m)} \pm \frac{1}{(s-2l)(\tilde{u}-2m)} \right] \quad (4.55)$$

is an ansatz which accounts for all $\log^2 U \log^2 V$ terms once we compare to $p_k(x, y)/(z - \bar{z})$ and solve for the coefficients. As an aside, [44] showed that the flavour projection can be undone in a way which gives the amplitude a remarkably simple form in terms of the "box diagram"

$$d^{I_1 I_2 I_3 I_4} = f^{J I_1 K} f^{K I_2 L} f^{L I_3 M} f^{M I_4 J}. \quad (4.56)$$

This derivation makes use of the identities

$$|G_F| (F_t)^0_a = 1, \quad d^{I_1 I_2 I_3 I_4} = \sum_a [G_F^\vee (F_t)^1_a]^2 P_a^{I_1 I_2 | I_3 I_4}. \quad (4.57)$$

If $k = 1$, it is straightforward to iterate the above procedure a few times to notice that

$$c_{lm} = \frac{4}{3} \left[ \frac{(l-1)^2}{l+m-2} + \frac{l^2 - 3l + 3}{l+m-3} - \frac{2(l-2)^2}{l+m-4} \right], \quad (4.58)$$

which agrees with the result of [44]. For all other values of $k$, we expect physically that

$$c_{lm}^{(k)} = \frac{1}{k}c_{lm}^{(1)} + d_{lm}^{(k)}, \tag{4.59}$$

where the remainder $d_{lm}^{(k)}$ (also symmetric under $l \leftrightarrow m$) becomes negligible in the flat space limit $l \sim m \to \infty$. We have only been able to find a general formula for $d_{lm}^{(k)}$ when the S-fold has $k = 2$. In this case, a workable method is to write

$$d_{lm} = \sum_{j=0}^{l-1} \frac{\mu_{l,j}}{l+m-j-2} \tag{4.60}$$

following [75] and guess $\mu_{l,j}$ as a function of $l$ for the first few values of $j$. It should come as no surprise that the result is proportional to $(l-j)_{j-1}$ which enforces the upper limit of the sum in (4.60). Once this is done, $\mu_{l,j}$ becomes a recognizable function up to a polynomial in $l$ which always has degree 6. Its coefficients can then be determined as functions of $j$ one-by-one. Explicitly,

$$\mu_{l,j} = \frac{(-1)^{j+l}(l-j)_{j-1}^2}{32(3)_{j-2}\left(l-j-\frac{3}{2}\right)_{j+2}}\Big[8(j-1)l^6 - 24(j-1)(j+2)l^5 + 12(2j^3 + 9j^2 - 10)l^4$$
$$- 8(j+2)(j^3 + 12j^2 - 10)l^3 + 2(18j^4 + 95j^3 + 82j^2 - 59j - 60)l^2$$
$$- 2(j+2)(23j^3 + 22j^2 - 11j - 12)l + 15j^4 + 33j^3 + 10j^2 - 14j - 8\Big] \tag{4.61}$$

leading to

$$\frac{(-1)^l}{2}d_{lm} = \frac{(l-1)^4 D_0(l,m)}{(2l-3)(2l-1)(l+m-2)} + \frac{(2l^6 - 12l^5 + 33l^4 - 58l^3 + 60l^2 - 45l + 11)D_1(l,m)}{(2l-5)(2l-3)(2l-1)(l+m-3)}$$
$$+ \frac{(l-2)^2(6l^4 - 42l^3 + 125l^2 - 184l + 107)D_2(l,m)}{(2l-7)(2l-5)(2l-3)(l+m-4)} \tag{4.62}$$
$$+ \frac{(l-3)^2(l-2)^2(6l^2 - 28l + 41)D_3(l,m)}{(2l-9)(2l-7)(2l-5)(l+m-5)} + \frac{2(l-4)^2(l-3)^2(l-2)^2 D_4(l,m)}{(2l-11)(2l-9)(2l-7)(l+m-6)},$$

where

$$D_n(l,m) \equiv {}_3F_2\left(1+n-l, 1+n-l, 2+n-l-m; \frac{5}{2}+n-l, 3+n-l-m; 1\right). \tag{4.63}$$

Mellin amplitudes of this type are appealing because they automatically match the subleading part of the double discontinuity referred to as $q(x,y)$ in (4.26). In order to demonstrate this, we will go to the crossed channel so that triple poles in $s$ produce $\log^2 V$ and triple poles in $t$ produce $\log^2 U$.[32] Within the $s = 2l$ residue however, there are also two sources of $\log U$. The first is the $t = 2m$ residue where we simply keep terms that were suppressed in (4.54). The second is an infinite sequence of double poles at even integers other than $t = 2m$ as dictated by the gamma functions. Keeping track of these leads to

$$\frac{\mathrm{dDisc}[H_a(U,V)]}{4\pi^2} = \frac{1}{\tilde{c}_{J,a}^2}\sum_{l,m=2}^\infty \frac{c_{lm}\Gamma(l+m-1)^2 U^m V^{l-2}}{16\Gamma(l-1)^2\Gamma(m-1)^2} \tag{4.64}$$
$$\left[\log^2 U + \left(\frac{\pm 2}{3-l-2m} - 4H_{m-2} + 4H_{l+m-2}\right)\log U\right]$$
$$+ \frac{1}{\tilde{c}_{J,a}^2}\sum_{l,m=2}^\infty\sum_{j\neq m}^\infty \frac{c_{lm}\Gamma(l+j-1)^2 U^j V^{l-2}}{8\Gamma(l-1)^2\Gamma(j-1)^2}\frac{j+l+m-3\pm(m-j)}{(j-m)(j+l+m-3)}\log U,$$

---

[32]Since both signs of (4.55) have the same dependence on $t$, even and odd spins will give rise to the same $\log^2 U \log^2 V$ term as expected.

where the upper (lower) sign is for even (odd) spins. Once the $k = 2$ coefficients given by (4.62) are plugged in, the above sum allows us to expand in $x$ and $y$ (after restoring the factor of $z - \bar{z}$) and precisely recover the $\log x$ terms seen in (4.29). In the last line of (4.64), this exercise fixes $j$ and $l$ to small integer values corresponding to the powers of $x$ and $y$ being examined. This causes all of the hypergeometric functions in (4.62) to simplify such that the remaining sum over $m$ can be done in closed form. Due to the ansatz (4.60) and the symmetry of $d_{lm}$, we can even evaluate the sum exactly when only finitely many $\mu_{l,j}$ residues are known. This is the case for the $k = 3$ and $k = 4$ S-folds. Solving for the subset $\{\mu_{2,j}, \ldots, \mu_{5,j}\}$, the truncated expressions (4.30) and (4.31) can all be matched confirming the absence of simple poles once again.

# 5 Conclusion

This work has been a quantitative exploration of holographic theories which are defined by an S-fold [30, 31, 35]. Part of it involved clarifying how basic features of the brane construction can be written as input to the conformal bootstrap. The AdS unitarity method [45] then led to interesting results, most importantly large-spin anomalous dimensions from table 7 and some finite-spin counterparts in (4.38) which can distinguish between different S-folds. Moreover, the algorithm we used in the process combined elements of [16–18] in a non-trivial way and eliminated the need to propose any sort of "alphabet" for the one-loop double discontinuity from the outset. Going up in twist three times was arbitrary and could have been done many more times.

It is natural to ask how much of the landscape of 4d $\mathcal{N} = 2$ holographic CFTs can be explored with these methods. Due to operator mixing, a single loop-level four-point function will continue to incorporate data from infinitely many tree-level four-point functions. In our case, these tree-level S-fold correlators, and the $k = 1$ correlators to which they were related, both obeyed a crucial constraint. The single-trace conformal primaries (in a consistent subsector) all had spins of at most $\mathcal{N}/2$ ensuring that none of them could come from long multiplets. For theories of class S, whose holographic duals were studied in [102], this will not be the case since the operators come from compactifying a more supersymmetric theory on a Riemann surface. Alternatively, SCFTs where the single-trace spins exceed $\mathcal{N}/2$ can still be tractable if they are related to maximally supersymmetric theories not through compactification but through a simpler orbifold procedure [103]. For S-folds in particular, we believe the following open problems should be investigated next.

- Double-trace anomalous dimensions, especially those in (4.38) which are reliable at low spin, can be helpful for interpreting numerical bootstrap results. In order for this application to proceed, it will be important to know the contributions from higher derivative terms in the effective action in addition to the loops. Noting that $\mathcal{S}_{D_4,1}^{(N)}$ has a marginal coupling, it is likely that these corrections depend more sensitively on the flavour group.

- Future targets include 4d $\mathcal{N} = 3$ SCFTs, for which the analogue of our OPE coefficient formula (2.42) is much richer. Since the desired correlators involve the $SU(3)$ polarizations mentioned in (1.2), projected tree-level exchanges no longer yield pure numbers but new harmonic polynomials. Limiting values of them should reduce to isoscalar factors which have been studied from a CFT perspective in [104].

- In contrast to the case of [35] which had the $\mathcal{N} = 2$ S-fold breaking global symmetries, $SU(4) \rightarrow SU(3)$ in $\mathcal{N} = 3$ S-folds is a breaking of R-symmetry. A $\frac{1}{2}$-BPS operator in $B\bar{B}[0;0]_p^{(0,p,0)}$ will therefore decompose into longer (but not long) multiplets. The superconformal blocks for these correlators are not known and indeed they might not even

be fixed kinematically. There is currently a gap in the literature concerning model dependent blocks which play a role in the mixing problems for model independent blocks.

- A decomposition along the lines of (2.20) exists for $\frac{1}{2}$-BPS operators of 4d $\mathcal{N} = 3$ theories [32] but the extra correlators that account for mixing will need to be treated without auxiliary functions. This motivates the need for a Lorentzian inversion formula which explicitly accounts for supersymmetry. Some initial work has been done in [105] leveraging dimension shifting identities for conformal blocks and it will be interesting to see how far this can be pushed.

We hope to begin addressing these in future work.

# Acknowledgements

I am grateful for discussions with Fernando Alday, Simone Giaocomelli, Tobias Hansen and Sakura Schäfer-Nameki during this work and for collaboration with Pietro Ferrero and Xinan Zhou which helped initiate it. Fernando Alday, Pietro Ferrero and Simone Giaocomelli gave helpful feedback on the draft. This project has received funding from the European Research Council (ERC) under the European Union's Horizon 2020 research and innovation programme (grant agreement No 787185).

# A    Standard inversion integrals

The inversion integrals needed to perform (2.31) were derived with elementary methods in [66]. They can also be seen as special cases of the crossing kernels found in [48]. To start, the appropriate double discontinuity whenever we have poles as $\bar{z} \to 1$ is

$$\mathrm{dDisc}\left[\left(\frac{1-\bar{z}}{\bar{z}}\right)^p\right] = 2\sin^2(\pi p)\left(\frac{1-\bar{z}}{\bar{z}}\right)^p. \tag{A.1}$$

Although (A.1) vanishes for $p \in \mathbb{Z}$, we can only rely on this for $p$ non-negative. When $p$ is a negative integer, the zero will be cancelled by a divergence in the $\bar{z}$ integral. Specifically,

$$\frac{r_{\bar{h}}^2}{4\pi^2}\int_0^1 \frac{d\bar{z}}{\bar{z}^2}\frac{k_{\bar{h}}(\bar{z})}{\bar{h}-\frac{1}{2}}\mathrm{dDisc}\left[\left(\frac{1-\bar{z}}{\bar{z}}\right)^p\right] = \frac{r_{\bar{h}}}{\Gamma(-p)^2}\frac{\Gamma(\bar{h}-p-1)}{\Gamma(\bar{h}+p+1)}. \tag{A.2}$$

In later parts of the calculation, a similar integral appears where the double discontinuity has already been taken.

$$r_{\bar{h}}^2\int_0^1 \frac{d\bar{z}}{\bar{z}^2}\frac{k_{\bar{h}}(\bar{z})}{\bar{h}-\frac{1}{2}}\left(\frac{1-\bar{z}}{\bar{z}}\right)^p = 2r_{\bar{h}}\Gamma(p+1)^2\frac{\Gamma(\bar{h}-p-1)}{\Gamma(\bar{h}+p+1)}. \tag{A.3}$$

The $z$ integral is also best handled by letting the exponent be generic and analytically continuing. A subtlety here is that the spectral density needs to be modified before its residues will give the OPE coefficients of operators with integer twists. As explained in [18], this is because the contour integral that recovers the OPE encircles certain poles of the shadow symmetric kernel in addition to those of the spectral density. Applying the necessary correction is equivalent to discarding all poles at values of $h$ that are independent of the integrand. This justifies

the last line of

$$\int_0^1 \frac{dz}{z^2} k_{1-h}(z) \left(\frac{z}{1-z}\right)^q = \frac{\Gamma(2-2h)\Gamma(1-q)^2\Gamma(q-h)}{\Gamma(1-h)^2\Gamma(2-q-h)}$$

$$\approx \pi \cot[\pi(q-h)] \frac{r_h}{\Gamma(q)^2} \frac{\Gamma(h+q-1)}{\Gamma(h-q+1)}. \tag{A.4}$$

It is then straightforward to obtain logarithmic versions of (A.4) through differentiation. The one we need is

$$\int_0^1 \frac{dz}{z^2} k_{1-h}(z) \left(\frac{z}{1-z}\right)^q \log\left(\frac{z}{1-z}\right) = \frac{\partial}{\partial q} \int_0^1 \frac{dz}{z^2} k_{1-h}(z) \left(\frac{z}{1-z}\right)^q$$

$$\approx -\pi^2 \csc^2[\pi(q-h)] \frac{r_h}{\Gamma(q)^2} \frac{\Gamma(h+q-1)}{\Gamma(h-q+1)} \tag{A.5}$$

up to terms with only simple poles. Mixed correlator generalizations of these results appear in the appendix of [86].

# B  Flavour crossing matrices

This appendix collects the explicit crossing matrices referred to in the main text. They are defined by

$$(F_t)^a_{\ b} = \frac{1}{\dim(\mathbf{R}_a)} P_a^{I_1 I_4 | I_3 I_2} P_b^{I_1 I_2 | I_3 I_4}, \quad (F_u)^a_{\ b} = \frac{1}{\dim(\mathbf{R}_a)} P_a^{I_1 I_3 | I_2 I_4} P_b^{I_1 I_2 | I_3 I_4}, \tag{B.1}$$

in terms of the flavour projectors in (2.29). For each flavour group in this list, we will state the S-fold where it appears and give the chosen ordering for the representations in terms of Dynkin labels. A subscript of $+$ will indicate the symmetric product of two adjoints and a subscript of $-$ will indicate the anti-symmetric product. This is enough to relate the two matrices as

$$(F_t)^a_{\ b} = (-1)^{|\mathbf{R}_a| + |\mathbf{R}_b|} (F_u)^a_{\ b} \tag{B.2}$$

so we only give $F_t$. In all cases, the singlet $(+)$ will have an index of 0 and the adjoint itself $(-)$ will have an index of 1. The matrix elements were obtained by converting the diagrammatic projectors in [106] back to index notation and performing the contraction with Cadabra [107–109].

$\underline{G_2}$, which can appear with $\mathcal{T}_{D_4,3}^{(N)}$, has

$$F_t = \begin{pmatrix} \frac{1}{14} & 1 & \frac{11}{12} & \frac{27}{14} & \frac{11}{2} \\ \frac{1}{14} & \frac{1}{2} & 0 & \frac{45}{56} & -\frac{11}{8} \\ \frac{1}{14} & 0 & \frac{1}{2} & -\frac{9}{28} & -\frac{1}{4} \\ \frac{1}{14} & \frac{5}{12} & -\frac{11}{12} & -\frac{29}{112} & \frac{11}{16} \\ \frac{1}{14} & -\frac{1}{4} & -\frac{1}{4} & \frac{27}{112} & \frac{3}{16} \end{pmatrix} \tag{B.3}$$

for the representations $[0,0]_+, [1,0]_-, [0,3]_-, [2,0]_+, [0,2]_+$.

$\underline{F_4}$, which can appear with $\mathcal{T}_{E_6,2}^{(N)}$, has

$$F_t = \begin{pmatrix} \frac{1}{52} & 1 & \frac{49}{2} & \frac{81}{13} & \frac{81}{4} \\ \frac{1}{52} & \frac{1}{2} & 0 & \frac{45}{26} & -\frac{9}{4} \\ \frac{1}{52} & 0 & \frac{1}{2} & -\frac{81}{91} & -\frac{9}{28} \\ \frac{1}{52} & \frac{5}{18} & -\frac{7}{9} & -\frac{7}{26} & \frac{3}{4} \\ \frac{1}{52} & -\frac{1}{9} & -\frac{7}{18} & \frac{3}{13} & \frac{1}{4} \end{pmatrix} \tag{B.4}$$

for the representations $[0,0,0,0]_+$, $[1,0,0,0]_-$, $[0,1,0,0]_-$, $[0,0,0,2]_+$, $[2,0,0,0]_+$.

$\underline{SU(2)}$, which can appear with $\mathcal{S}_{A_2,2}^{(N)}$, $\mathcal{S}_{D_4,2}^{(N)}$, $\mathcal{S}_{A_2,4}^{(N)}$, $\mathcal{T}_{A_1,3}^{(N)}$ and $\mathcal{T}_{A_2,4}^{(N)}$, has

$$F_t = \begin{pmatrix} \frac{1}{3} & 1 & \frac{5}{3} \\ \frac{1}{3} & \frac{1}{2} & -\frac{5}{6} \\ \frac{1}{3} & -\frac{1}{2} & \frac{1}{6} \end{pmatrix} \tag{B.5}$$

for the representations $[0]_+, [2]_-, [4]_+$.

$\underline{SU(n)}$ with $n > 2$, which can appear with $\mathcal{S}_{D_4,3}^{(N)}$ and $\mathcal{T}_{A_2,2}^{(N)}$, has

$$F_t = \begin{pmatrix} \frac{1}{n^2-1} & 1 & \frac{n^2(n-3)}{4(n-1)} & \frac{n^2(n+3)}{4(n+1)} & 1 & \frac{\sqrt{2}}{4}(n^2-4) \\ \frac{1}{n^2-1} & \frac{1}{2} & \frac{n(n-3)}{4(n-1)} & -\frac{n(n+3)}{4(n+1)} & \frac{1}{2} & 0 \\ \frac{1}{n^2-1} & \frac{1}{n} & \frac{n^2-n+2}{4(n-1)(n-2)} & \frac{1}{4}\frac{n+3}{n+1} & -\frac{1}{n-2} & -\frac{\sqrt{2}}{4}\frac{n+2}{n} \\ \frac{1}{n^2-1} & -\frac{1}{n} & \frac{1}{4}\frac{n-3}{n-1} & \frac{n^2+n+2}{4(n+1)(n+2)} & \frac{1}{n+2} & -\frac{\sqrt{2}}{4}\frac{n-2}{n} \\ \frac{1}{n^2-1} & \frac{1}{2} & -\frac{n^2(n-3)}{4(n-1)(n-2)} & \frac{n^2(n+3)}{4(n+1)(n+2)} & \frac{1}{2}\frac{n^2-12}{n^2-4} & -\frac{\sqrt{2}}{2} \\ \frac{\sqrt{2}}{n^2-1} & 0 & -\frac{\sqrt{2}}{4}\frac{n(n-3)}{(n-1)(n-2)} & -\frac{\sqrt{2}}{4}\frac{n(n+3)}{(n+1)(n+2)} & -\frac{2\sqrt{2}}{n^2-4} & \frac{1}{2} \end{pmatrix} \tag{B.6}$$

for the representations $[0,\dots,0]_+$, $[1,0,\dots,0,1]_-$, $[0,1,0,\dots,0,1,0]_+$, $[2,0,\dots,0,2]_+$, $[1,0,\dots,0,1]_+$, $[2,0,\dots,0,1,0]_- \oplus [0,1,0,\dots,0,2]_-$.

$\underline{SO(n)}$ with $n > 2$, which can appear with $\mathcal{T}_{D_4,2}^{(N)}$, has

$$F_t = \begin{pmatrix} \frac{2}{n(n-1)} & \frac{n+2}{n} & \frac{(n-3)(n+2)}{6} & \frac{(n-2)(n-3)}{12} & 1 & \frac{(n-3)(n+2)}{4} \\ \frac{2}{n(n-1)} & \frac{n^2-8}{2n(n-2)} & \frac{(n-4)(n-3)(n+1)}{6(n-1)(n-2)} & -\frac{n-3}{6} & \frac{1}{2}\frac{n-4}{n-2} & -\frac{n-3}{n-2} \\ \frac{2}{n(n-1)} & \frac{n-4}{n(n-2)} & \frac{n^2-6n+11}{3(n-1)(n-2)} & \frac{1}{6} & -\frac{1}{n-2} & -\frac{1}{2}\frac{n-4}{n-2} \\ \frac{2}{n(n-1)} & -2\frac{n+2}{n(n-2)} & \frac{(n+1)*(n+2)}{3(n-1)(n-2)} & \frac{1}{6} & \frac{2}{n-2} & -\frac{1}{2}\frac{n+2}{n-2} \\ \frac{2}{n(n-1)} & \frac{(n-4)(n+2)}{2n(n-2)} & -\frac{(n-3)(n+1)(n+2)}{6(n-1)(n-2)} & \frac{n-3}{6} & \frac{1}{2} & 0 \\ \frac{2}{n(n-1)} & -\frac{4}{n(n-2)} & -\frac{(n-4)(n+1)}{3(n-1)(n-2)} & -\frac{1}{6} & 0 & \frac{1}{2} \end{pmatrix} \tag{B.7}$$

for the representations $[0,\dots,0]_+$, $[0,1,0,\dots,0]_-$, $[2,0,\dots,0]_+$, $[0,2,0,\dots,0]_+$, $[0,0,0,1,0,\dots,0]_+$, $[1,0,1,0,\dots,0]_-$.

$\underline{USp(n)}$ with $n > 2$, which can appear with $\mathcal{S}_{D_4,2}^{(N)}$ and $\mathcal{S}_{E_6,2}^{(N)}$, has $F_t$ given by (B.7) with $n \mapsto -n$ [106].

## C  Improvements to the resummation

### C.1  Simpler differential operators

Our main results were encoded in a double expansion for the one-loop integrand of (2.31). Powers of $x$ and $y$ respectively allowed one to go up in twist and down in spin. While (4.24) solved the problem of computing arbitrarily many terms in principle, the growing number of Casimir operators led to a significant computational cost. Fortunately, (4.18) shows that a single pole in $\bar{h}(\bar{h}-1)$ was responsible for almost all of the Casimirs we had to introduce. Moving it to a more natural place can be done with telescopic identities which were found to

simplify loop calculations in [66]. The general form they take is

$$B(n,\ell) = \sum_{2j=0}^{n} \frac{f_k(2j)(2j+1)^2 \bar{h}(\bar{h}-1)}{\bar{h}(\bar{h}-1) - (n+1)(n+2)} R_{-2j-2}(n+2)R_{2j}(\bar{h}) \tag{C.1}$$

$$= \sum_{2j=0}^{n} T_k(2j)\bar{h}(\bar{h}-1)R_{-2j-2}(n+2)R_{2j+1}(\bar{h}),$$

where $f_k(2j)$ is a kernel implementing $2m_L|k$ and $k \in \{1,2,3,4\}$. Solving for

$$T_k(2j) = \sum_{l=0}^{2j} f_k(l)(l+1)^2, \tag{C.2}$$

it is then straightforward to refer to table 5 and arrive at the explicit expressions which were given in (4.53). The sums involving $T_k(2j)$ instead of $f_k(2j)$ are advantageous because they can be brought into the form assumed by [17] using only a single quadratic Casimir. The function on which it acts will be one of

$$\frac{1}{2}\sum_{\ell=0}^{\infty} R_{2j+1}(h_0+\ell)k_{h_0+\ell}\left(\frac{1}{x+1}\right) = \lim_{\eta\to2j+1}\Gamma(-\eta)^2\left[\frac{1}{2}x^\eta + \sum_{m=0}^{\infty}\frac{\partial}{\partial m}\frac{\mathcal{A}^+_{\eta,-m-1}(h_0)}{2x^{-m}}\right]$$

$$\frac{1}{2}\sum_{\ell=0}^{\infty}(-1)^\ell R_{2j+1}(h_0+\ell)k_{h_0+\ell}\left(\frac{1}{x+1}\right) = \lim_{\eta\to2j+1}\Gamma(-\eta)^2\sum_{m=0}^{\infty}\frac{\partial}{\partial m}\frac{\mathcal{A}^-_{\eta,-m-1}(h_0)}{2x^{-m}}. \tag{C.3}$$

These are no more difficult to evaluate than (4.23). Going through the same steps as before, it becomes clear that

$$H_n^J(x,y) = \frac{x^2}{y^2}\frac{(2n+3)!}{(n+1)!^2}P_{n+1}(2x+1)\log x \sum_{\ell^\pm} B(n,\ell)(-1)^{n+\ell}k_{n+\ell+3}(-y) - \frac{x^2}{y^2}(-1)^n k_{n+2}(-y)$$

$$\sum_{2j=0}^{n} T_k(2j)R_{-2j-2}(n+2)\mathcal{D}\left[\frac{x^{2j+1}\log x(\log x + 2H_{2j+n+3} + 2H_{n-2j} - 4H_{2j+1})}{4(2j+1)!^2}\right]$$

$$-\frac{(n-2j)!}{2}\sum_{m=0}^{\infty}\frac{x^m\log x}{m!^2}\left(\frac{(n+2)_{m+1}(-n-1)_m}{(-1)^m(2j+n+3)!}\frac{1-\delta_{m,2j+1}}{2j+1-m}\right.$$

$$\left.\pm\sum_{l=0}^{m}\frac{(2j+2)_l(2j+2-m)_l}{l!\Gamma(2j+n+4-m+l)}\frac{(-m)_l(n-2j+1)_m}{(2j-n-m)_l}\right)\right] + \dots \tag{C.4}$$

is the desired more efficient version of (4.24).

## C.2 One less infinite sum

Paying attention to the Pochhammer symbols in (C.4), there is a single infinite sum associated with the second line of (C.3). Although it has not been necessary here, one can rewrite it in a nice way by noticing that

$$\mathcal{A}^-_{\eta,-m-1}(h_0) = -\frac{\Gamma(h_0+m-\eta-1)}{m!^2\Gamma(-\eta)^2\Gamma(\eta-m+h_0)}\,{}_3F_2\left[\begin{array}{ccc}-m, & \eta+1, & \eta+1-m\\ \eta-m+h_0, & \eta+2-m-h_0\end{array}\right]$$

$$= \frac{(-1)^{m+1}}{m!\Gamma(\eta+1)\Gamma(-\eta)^2}\frac{\Gamma(h_0+m-\eta-1)}{(\eta+2-m-h_0)_{h_0-1}}. \tag{C.5}$$

The first line holds generally while the second is a zero-balanced summation formula from [110] valid for $m > h_0 - 2 = n + 1$. A simple consequence is that

$$\sum_{m=n+2}^{\infty} \sum_{l=0}^{m} \frac{(2j+2)_l (2j+2-m)_l (-m)_l}{(2j-n-m)_l (n-2j+m+1)_{4j+3+l}} \frac{x^m}{l! m!^2} = \frac{(-x)^{n+2} \Gamma(2n-2j+3)}{(2j+1)!(n+2)!(2j-2n-2)_{n+2}}$$
$$_2F_1(1, 1+n-2j; n+3; -x). \qquad (C.6)$$

To rewrite the original expression (4.24) in the same way, one can just take $2j \mapsto n$ above. Although (C.6) represents a certain contribution to the double discontinuity at one loop, it bears a striking similarity to the tree-level result (4.64). This is perhaps an indication that the present algorithm will be useful for computing the OPE coefficient corrections $\langle a^{(2)} \rangle_{a,n,\ell}$.

## C.3 Generalization to arbitrary dimensions

In the case of 4d CFTs, (4.24) and (C.4) have provided a successful starting point for analyzing

$$G_a^J(U, V) = \sum_{n=0}^{\infty} \sum_{\ell} \frac{1}{8} \langle a^{(0)} \gamma^{(1)2} \rangle_{a,n,\ell} \frac{(z-\bar{z})U^2}{V^3} g_{2\Delta_\phi + 2n + \ell, \ell}(V, U) \log^2 V. \qquad (C.7)$$

Since this function is very similar to the one expected for the double discontinuity in all dimensions, it will be beneficial to find expansions of it which do not rely on the conformal blocks having a closed form. To this end, let us consider

$$\widetilde{G}^J(U, V) = \sum_{n=0}^{\infty} \sum_{\ell} B(n, \ell) g_{2\Delta_\phi + 2n + \ell, \ell}(V, U) \equiv \sum_{n=0}^{\infty} \widetilde{H}_n^J(U, V), \qquad (C.8)$$

which is simply the desired function with cross ratios stripped off. The procedure for extracting OPE data via the inversion formula relies on the $\log^2 U$, $\log U$ and regular terms in the $s$-channel lightcone limit. Once these are known, it is straightforward to re-expand in $x$ and $y$. A strategy for finding them without explicit conformal blocks is to first consider the $t$-channel lightcone limit which permits

$$g_{\tau, \ell}(V, U) = \left( \frac{V}{1-U} \right)^{\frac{\tau}{2}} \left[ k_{\frac{\tau}{2} + \ell}(1-U) + O(V) \right] \qquad (C.9)$$

to be used. Techniques from [17] then become applicable and lead to

$$\widetilde{H}_n^J(U, V) = V^{\Delta_\phi + n} \left[ \sum_{l=0}^{\infty} U^l \left( \log^2 U \, \alpha_{l,0}^{(n)} + \log U \, \beta_{l,0}^{(n)} + \gamma_{l,0}^{(n)} \right) + O(V) \right]. \qquad (C.10)$$

The idea is then to regard (C.10) as a boundary condition and study the $s$-channel with a differential equation.[33] This is possible because the $\widetilde{H}_n^J(U, V)$ functions are examples of *twist conformal blocks* — linear combinations of conformal blocks that differ in spin but have the same twist. As shown in [16], these all satisfy the same differential equation found by combining the quadratic and quartic Casimirs such that $\ell$ drops out of the eigenvalue. This equation is

$$\left[ \mathcal{D}_4 - \mathcal{D}_2^2 + [d^2 - d(2\tau + 3) + \tau^2 + 2\tau + 2] \mathcal{D}_2 - \lambda_\tau \right] \widetilde{H}_{\frac{\tau}{2} - \Delta_\phi}^J(U, V) = 0$$
$$\lambda_\tau = \frac{\tau}{4} [d^2(5\tau + 6) - 2d(2\tau^2 + 5\tau + 2) + \tau(\tau + 2)^2 - 2d^3], \qquad (C.11)$$

---

[33]This makes it important to work with the cross ratios $U$ and $V$. With $z$ and $\bar{z}$ there would be spurious solutions which are not symmetric under $z \longleftrightarrow \bar{z}$. A simple example is the one obtained by replacing all 4d blocks with only one of the two terms in (2.24).

where

$$\mathcal{D}_2 = \mathcal{D} + \bar{\mathcal{D}} + (d-2)\frac{z\bar{z}}{z-\bar{z}}\left[(1-z)\frac{\partial}{\partial z} - (1-\bar{z})\frac{\partial}{\partial\bar{z}}\right]$$

$$\mathcal{D}_4 = \left(\frac{z\bar{z}}{z-\bar{z}}\right)^{d-2}(\mathcal{D}-\bar{\mathcal{D}})\left(\frac{z\bar{z}}{z-\bar{z}}\right)^{2-d}(\mathcal{D}-\bar{\mathcal{D}}) \tag{C.12}$$

referring to (4.18). The approach of [16] then consists of applying (C.11) to

$$\widetilde{H}_n^J(U,V) = V^{\Delta_\phi+n}\sum_{l=0}^{\infty}\sum_{m=0}^{\infty}U^l V^m\left(\log^2 U\,\alpha_{l,m}^{(n)} + \log U\,\beta_{l,m}^{(n)} + \gamma_{l,m}^{(n)}\right), \tag{C.13}$$

in order to solve for $\left\{\alpha_{l,m}^{(n)},\beta_{l,m}^{(n)},\gamma_{l,m}^{(n)}\right\}$ in terms of $\left\{\alpha_{l,m-1}^{(n)},\beta_{l,m-1}^{(n)},\gamma_{l,m-1}^{(n)}\right\}$ which are eventually known due to (C.10).

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
