# Peer review of "Holographic S-fold theories at one loop"

_SciPost Physics, doi:SciPost Phys. 12, 149 (2022)_

## Round 1 · Referee Report · Anonymous · 2022-2-28

Report
This paper proposes to compute various specific 4d N=2 theories at 1-loop, generalizing previous studies. The theories considered differ at tree level just by the spectrum of allowed operators, so at 1-loop the holographic correlator calculation differs just by restricting the sum over double trace operators. The basic idea is basically identical to previous studies with maximal supersymmetry and orbifolds (where again 1-loop was distinguished by restricting the sum), except now applied to theories with less supersymmetry.
The main problem is that the 1-loop calculation seems be incomplete. While the double discontinuity was computed in an expansion (and given in 4.26), it was not completed to the full 1-loop correlator (up to contact terms of course). There are by now many standard ways of doing this step in the literature (Mellin space is often the easiest), and I think this step needs to be completed before this draft is ready for publication.
On a related note, the 1-loop data has so far only been computed in a large spin expansion, whereas it should be possible to compute it at finite spin, as has been done in all previous cases by now.
Author: Connor Behan on 2022-03-17 [id 2295]
(in reply to Report 1 on 2022-02-28)Thank you for responding so quickly. On the arXiv, I have added sections 4.3.3 and 4.5 which guess some finite-spin anomalous dimensions and the Mellin amplitude respectively by extrapolating the $k = 2$ coefficients. Neither calculation appears to work for the $k = 3, 4$ S-folds which have no analogue with maximal SUSY.
The problem with using the projection method of ref [91] for these cases is that we cannot numerically approximate the Mellin amplitude by cutting off a divergent sum. If I focus on just the correction term for $k = 2$ (with only the softer $d_{lm}$ coefficients and no $c_{lm}$), the very lowest CFT data starts to look right up to a factor of 2 which got lost somewhere. But I would need to understand the first issue much better before I could stand behind a result like this.
Anonymous on 2022-03-20 [id 2305]
(in reply to Connor Behan on 2022-03-17 [id 2295])I am not able to see the updated pdf on Arxiv for some reason (I don't see the new sections).
Anonymous on 2022-03-21 [id 2309]
(in reply to Anonymous Comment on 2022-03-20 [id 2305])You're right. I'm not sure what happened. In any case, v3 is now ready and it should appear in two days.

---

## Round 1 · Referee Report · Anonymous · 2022-4-7

Report
In this paper the author studies four-point functions of Higgs branch operators for N=2 S-fold theories at one loop using the AdS unitarity method. This analysis generalizes previous one loop results for N=4 SYM and tree level computations for N=2 SCFTs from D3-branes probing F-theory singularities (due to flat 7-branes).
The one loop computation is performed in an asymptotic expansion around large spin and the problem of evaluating anomalous dimensions at finite spin is addressed for a subclass of S-fold theories: A closed form expression valid at fixed integer spin is provided for S-fold models with twist of order two.
The results are clearly presented and in my opinion this work constitutes an interesting starting point for future studies of holographic correlators for SCFTs with non-maximal superconformal symmetry. I therefore recommend this paper for publication in SciPost Physics.
Even though the author focuses on a specific class of models, it seems that the range of applicability of the techniques used in this paper is wider. In particular it seems that the only crucial assumption is that the super descendants of the operators under consideration have at most spin 1. I think it would be helpful if the author, prior to publication, explained more in detail what are the limitations of the method and whether it can be applied to other holographic N=2 superconformal theories, such as class S models.

---

## Round 4 · Referee Report · Anonymous (Referee 1) · 2022-4-12

Report

With the new corrections, I think the draft is ready for submission. I think it would be nice if the $k>2$ case could also be given an exact expression, but perhaps that will require new methods.

---

## Round 4 · List of Changes

In accordance with the referee comments: 1. Section 4.3.3 has been added to report spin-1 and spin-2 anomalous dimensions for the k=2 S-fold. 2. Section 4.5 has been added to discuss the one-loop Mellin amplitude. 3. A paragraph has been added to section 5 about which complications might arise in other theories.

Additionally: 4. There are now a few more citations to prior work on holographic correlators. 5. The main results previously had a sign error and a more substantial coefficient error in the k=4 case. These have been corrected.

---

## Editorial Decision

published